# Scalable and Order-robust Continual Learning with Additive Parameter Decomposition

**Jaehong Yoon[1], Saehoon Kim[2], Eunho Yang[1,2], and Sung Ju Hwang[1,2]**
KAIST[1], AITRICS[2], South Korea
`{jaehong.yoon, eunhoy, sjhwang82}@kaist.ac.kr, shkim@aitrics.com`

## Abstract

While recent continual learning methods largely alleviate the catastrophic problem on toy-sized datasets, some issues remain to be tackled to apply them to real-world problem domains. First, a continual learning model should effectively handle catastrophic forgetting and be efficient to train even with a large number of tasks. Secondly, it needs to tackle the problem of *order-sensitivity*, where the performance of the tasks largely varies based on the order of the task arrival sequence, as it may cause serious problems where fairness plays a critical role (e.g. medical diagnosis). To tackle these practical challenges, we propose a novel continual learning method that is scalable as well as order-robust, which instead of learning a completely shared set of weights, represents the parameters for each task as a sum of task-shared and *sparse* task-adaptive parameters. With our *Additive Parameter Decomposition (APD)*, the task-adaptive parameters for earlier tasks remain mostly unaffected, where we update them only to reflect the changes made to the task-shared parameters. This decomposition of parameters effectively prevents catastrophic forgetting and order-sensitivity, while being computation- and memory-efficient. Further, we can achieve even better scalability with APD using *hierarchical knowledge consolidation*, which clusters the task-adaptive parameters to obtain hierarchically shared parameters. We validate our network with APD, *APD-Net*, on multiple benchmark datasets against state-of-the-art continual learning methods, which it largely outperforms in accuracy, scalability, and order-robustness.

## 1 Introduction

Continual learning (Thrun, 1995), or lifelong learning, is a learning scenario where a model is incrementally updated over a sequence of tasks, potentially performing knowledge transfer from earlier tasks to later ones. Building a successful continual learning model may lead us one step further towards developing a general artificial intelligence, since learning numerous tasks over a long-term time period is an important aspect of human intelligence. Continual learning is often formulated as an incremental / online multi-task learning that models complex task-to-task relationships, either by sharing basis vectors in linear models (Kumar & Daume III, 2012; Ruvolo & Eaton, 2013) or weights in neural networks (Li & Hoiem, 2016). One problem that arises here is that as the model learns on the new tasks, it could forget what it learned for the earlier tasks, which is known as the problem of *catastrophic forgetting*. Many recent works in continual learning of deep networks (Li & Hoiem, 2016; Lee et al., 2017; Shin et al., 2017; Kirkpatrick et al., 2017; Riemer et al., 2019; Chaudhry et al., 2019) tackle this problem by introducing advanced regularizations to prevent drastic change of network weights. Yet, when the model should adapt to a large number of tasks, the interference between task-specific knowledge is inevitable with fixed network capacity. Recently introduced expansion-based approaches handle this problem by expanding the network capacity as they adapt to new tasks (Rusu et al., 2016; Fang et al., 2017; Yoon et al., 2018; Li et al., 2019). These recent advances have largely alleviated the catastrophic forgetting, at least with a small number of tasks.

However, to deploy continual learning to real-world systems, there are a number of issues that should be resolved. First, in practical scenarios, the number of tasks that the model should train on may be large. In the lifelong learning setting, the model may even have to continuously train on an unlimited number of tasks. Yet, conventional continual learning methods have not been verified for their scalability to a large number of tasks, both in terms of effectiveness in the prevention of

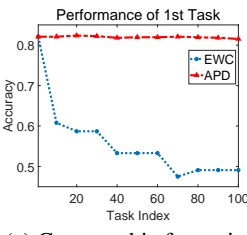
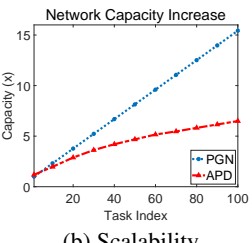
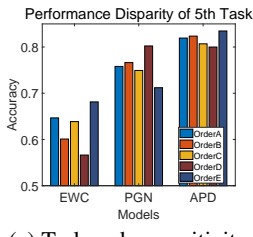

(a) Catastrophic forgetting     (b) Scalability     (c) Task-order sensitivity

Figure 1: Description of crucial challenges for continual learning with Omniglot dataset experiment. **Catastrophic forgetting:** Model should not forget what it has learned about previous tasks. **Scalability:** The increase in network capacity with respect to the number of tasks should be minimized. **Order sensitivity:** The model should have similar final performance regardless of the task order. Our model with Additive Parameter Decomposition effectively solves these three problems.

catastrophic forgetting, and efficiency as to memory usage and computations (See Figure 1 (a), and (b)).

Another important but relatively less explored problem is the problem of *task order sensitivity*, which describes the performance discrepancy with respect to the task arrival sequence (See Figure 1 (c)). The task order that the model trains on has a large impact on the individual task performance as well as the final performance, not only because of the model drift coming from the catastrophic forgetting but due to the unidirectional knowledge transfer from earlier tasks to later ones. This order-sensitivity could be highly problematic if fairness across tasks is important (e.g. disease diagnosis).

To handle these practical challenges, we propose a novel continual learning model with *Additive Parameter Decomposition (APD)*. APD decomposes the network parameters at each layer of the target network into task-shared and sparse task-specific parameters with small mask vectors. At each arrival of a task to a network with APD, which we refer to as *APD-Net*, it will try to maximally utilize the task-shared parameters and will learn the incremental difference that cannot be explained by the shared parameters using sparse task-adaptive parameters. Moreover, since having a single set of shared parameters may not effectively utilize the varying degree of knowledge sharing structure among the tasks, we further cluster the task-adaptive parameters to obtain hierarchically shared parameters (See Figure 2).

This decomposition of generic and task-specific knowledge has clear advantages in tackling the previously mentioned problems. First, APD will largely alleviate catastrophic forgetting, since learning on later tasks will have no effect on the task-adaptive parameters for the previous tasks, and will update the task-shared parameters only with generic knowledge. Secondly, since APD does not change the network topology as existing expansion-based approaches do, APD-Net is memory-efficient, and even more so with hierarchically shared parameters. It also trains fast since it does not require multiple rounds of retraining. Moreover, it is order-robust since the task-shared parameters can stay relatively static and will converge to a solution rather than drift away upon the arrival of each task. With the additional mechanism to retroactively update task-adaptive parameters, it can further alleviate the order-sensitivity from unidirectional knowledge transfer as well.

We validate our methods on several benchmark datasets for continual learning while comparing against state-of-the-art continual learning methods to obtain significantly superior performance with minimal increase in network capacity while being scalable and order-robust.

The contribution of this paper is threefold:

- We tackle practically important and novel problems in continual learning that have been overlooked thus far, such as *scalability* and *order robustness*.

- We introduce a novel framework for continual deep learning that effectively prevents catastrophic forgetting, and is highly scalable and order-robust, which is based on the decomposition of the network parameters into shared and sparse task-adaptive parameters with small mask vectors.

- We perform extensive experimental validation of our model on multiple datasets against recent continual learning methods, whose results show that our method is significantly superior to them in terms of the accuracy, efficiency, scalability, as well as order-robustness.

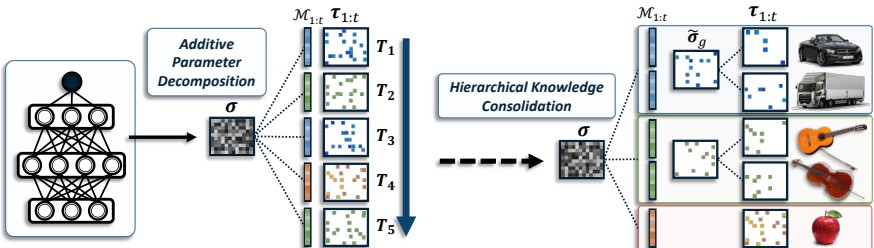

Figure 2: An illustration of Additive Parameter Decomposition (APD) for continual learning. APD effectively prevents catastrophic forgetting and suppresses order-sensitivity by decomposing the model parameters into shared $\sigma$ and sparse task-adaptive $\tau_t$, which will let later tasks to only update shared knowledge. $\mathcal{M}_t$ is the task-adaptive mask on $\sigma$ to access only the relevant knowledge. Sparsity on $\tau_t$ and hierarchical knowledge consolidation which hierarchically rearranges the shared parameters greatly enhances scalability.

## 2 RELATED WORK

**Continual Learning** The literature on continual (lifelong) learning (Thrun, 1995) is vast (Ruvolo & Eaton, 2013) as it is a long-studied topic, but we only mention the most recent and relevant works. Most continual deep learning approaches are focused on preventing *catastrophic forgetting*, in which case the retraining of the network for new tasks shifts the distribution of the learned representations. A simple yet effective regularization is to enforce the representations learned at the current task to be closer to ones from the network trained on previous tasks (Li & Hoiem, 2016). A more advanced approach is to employ deep generative models to compactly encode task knowledge (Shin et al., 2017) and generate samples from the model later when learning for a novel task. Kirkpatrick et al. (2017), and Schwarz et al. (2018) proposed to regularize the model parameter for the current tasks with parameters for the previous task via a Fisher information matrix, to find a solution that works well for both tasks, and Lee et al. (2017) introduces a moment-matching technique with a similar objective. Serrà et al. (2018) proposes a new binary masking approach to minimize drift for important prior knowledge. The model learns pseudo-step function to promote hard attention, then builds a compact network with a marginal forgetting. But the model cannot expand the network capacity and performs unidirectional knowledge transfer thus suffers from the order-sensitivity. Lopez-Paz & Ranzato (2017); Chaudhry et al. (2019) introduces a novel approach for efficient continual learning with weighted update according to the gradients of episodic memory under single-epoch learning scenario. Nguyen et al. (2018) formulates continual learning as a sequential Bayesian update and use coresets, which contain important samples for each observed task to mitigate forgetting when estimating the posterior distribution over weights for the new task. Riemer et al. (2019) addresses the stability-plasticity dilemma maximizing knowledge transfer to later tasks while minimizing their interference on earlier tasks, using optimization-based meta-learning with experience replay.

**Dynamic Network Expansion** Even with well-defined regularizers, it is nearly impossible to completely avoid catastrophic forgetting, since in practice, the model may encounter an unlimited number of tasks. An effective way to tackle this challenge is by dynamically expanding the network capacity to handle new tasks. Dynamic network expansion approaches have been introduced in earlier work such as Zhou et al. (2012), which proposed an iterative algorithm to train a denoising autoencoder while adding in new neurons one by one and merging similar units. Rusu et al. (2016) proposed to expand the network by augmenting each layer of a network by a fixed number of neurons for each task, while keeping the old weights fixed to avoid catastrophic forgetting. Yet, this approach often results in a network with excessive size. Yoon et al. (2018) proposed to overcome these limitations via selective retraining of the old network while expanding each of its layer with only the necessary number of neurons, and further alleviate catastrophic forgetting by splitting and duplicating the neurons. Xu & Zhu (2018) proposed to use reinforcement learning to decide how many neurons to add. Li et al. (2019) proposes to perform an explicit network architecture search to decide how much to reuse the existing network weights and how much to add. Our model also performs dynamic network expansion as the previous expansion-based methods, but instead of adding in new units, it additively decomposes the network parameters into task-shared and task-specific parameters. Further, the capacity increase at the arrival of each task is kept minimal with the sparsity on the task-specific parameters and the growth is logarithmic with the hierarchical structuring of shared parameters.

## 3 CONTINUAL LEARNING WITH ADDITIVE PARAMETER DECOMPOSITION

In a continual learning setting, we assume that we have sequence of tasks $\{\mathcal{T}_1, \ldots, \mathcal{T}_T\}$ arriving to a deep network in a random order. We denote the dataset of the $t^{th}$ task as $\mathcal{D}_t = \{\mathbf{x}_t^i, \mathbf{y}_t^i\}_{i=1}^{N_t}$, where $\mathbf{x}_t^i$ and $\mathbf{y}_t^i$ are $i^{th}$ instance and label among $N_t$ examples. We further assume that they become inaccessible after step $t$. The set of parameters for the network at step $t$ is then given as $\mathbf{\Theta}_t = \{\boldsymbol{\theta}_t^l\}$, where $\{\boldsymbol{\theta}_t^l\}$ represents the set of weights for each layer $l$; we omit the layer index $l$ when the context is clear. Then the training objective at the arrival of task $t$ can be defined as follows: minimize$_{\mathbf{\Theta}_t}$ $\mathcal{L}(\mathbf{\Theta}_t; \mathbf{\Theta}_{t-1}, \mathcal{D}_t) + \lambda \mathcal{R}(\mathbf{\Theta}_t)$, where $\mathcal{R}(\cdot)$ is a regularization term on the model parameters. In the next paragraph, we introduce our continual learning framework with task-adaptive parameter decomposition and hierarchical knowledge consolidation.

**Additive Parameter Decomposition**    To minimize the effect of catastrophic forgetting and the amount of newly introduced parameters with network expansion, we propose to decompose $\boldsymbol{\theta}$ into a *task-shared* parameter matrix $\boldsymbol{\sigma}$ and a *task-adaptive* parameter matrix $\boldsymbol{\tau}$, that is, $\boldsymbol{\theta}_t = \boldsymbol{\sigma} \otimes \mathcal{M}_t + \boldsymbol{\tau}_t$ for task $t$, where the masking variable $\mathcal{M}_t$ acts as an *attention* on the task-shared parameter to guide the learner to focus only on the parts relevant for each task. This decomposition allows us to easily control the trade-off between semantic drift and predictive performance of a new task by imposing separate regularizations on decomposed parameters. When a new task arrives, we encourage the shared parameters $\boldsymbol{\sigma}$ to be properly updated, but not deviate far from the previous shared parameters $\boldsymbol{\sigma}^{(t-1)}$. At the same time, we enforce the capacity of $\boldsymbol{\tau}_t$ to be as small as possible, by making it sparse. The objective function for this decomposed parameter model is given as follows:

$$\underset{\boldsymbol{\sigma}, \boldsymbol{\tau}_t, \mathbf{v}_t}{\text{minimize}} \quad \mathcal{L}\left(\{\boldsymbol{\sigma} \otimes \mathcal{M}_t + \boldsymbol{\tau}_t\}; \mathcal{D}_t\right) + \lambda_1 \|\boldsymbol{\tau}_t\|_1 + \lambda_2 \|\boldsymbol{\sigma} - \boldsymbol{\sigma}^{(t-1)}\|_2^2, \tag{1}$$

where $\mathcal{L}$ denotes a loss function, $\boldsymbol{\sigma}^{(t-1)}$ denotes the shared parameter before the arrival of the current task $t$, $\|\cdot\|_1$ indicates an element-wise $\ell_1$ norm defined on the matrix, and $\lambda_1$, $\lambda_2$ are hyperparameters balancing efficiency catastrophic forgetting. We use $\ell_2$ transfer regularization to prevent catastrophic forgetting, but we could use other types of regularizations as well, such as Elastic Weight Consolidation (Kirkpatrick et al., 2017). The masking variable $\mathcal{M}_t$ is a sigmoid function with a learnable parameter $\mathbf{v}_t$, which is applied to output channels or neurons of $\boldsymbol{\sigma}$ in each layer. We name our model with decomposed network parameters, *Additive Parameter Decomposition (APD)*.

The proposed decomposition in (1) makes continual learning efficient, since at each task we only need to learn a very sparse $\boldsymbol{\tau}_t$ that accounts for task-specific knowledge that cannot be explained with the transformed shared knowledge $\boldsymbol{\sigma} \otimes \mathcal{M}_t$. Thus, in a way, we are doing residual learning with $\boldsymbol{\tau}_t$. Further, it helps the model achieve robustness to the task arrival order, because semantic drift occurs only through the task-shared parameter that corresponds to *generic* knowledge, while the *task-specific* knowledge learned from previous tasks are kept intact. In the next section, we introduce additional techniques to achieve even more task-order robustness and efficiency.

**Order Robust Continual Learning with Retroactive Parameter Updates**    We observe that a naive update of the shared parameters may induce semantic drift in parameters for the previously trained tasks which will yield an order-sensitive model, since we do not have access to previous task data. In order to provide high degree of order-robustness, we impose an additional regularization to further prevent *parameter-level* drift without explicitly training on the previous tasks.

To achieve order-robustness in (1), we need to *retroactively* update task adaptive parameters of the past tasks to reflect the updates in the shared parameters at each training step, so that all previous tasks are able to maintain their original solutions. Toward this objective, when a new task $t$ arrives, we first recover all previous parameters ($\boldsymbol{\theta}_i$ for task $i < t$): $\boldsymbol{\theta}_i^* = \boldsymbol{\sigma}^{(t-1)} \otimes \mathcal{M}_i^{(t-1)} + \boldsymbol{\tau}_i^{(t-1)}$ and then update $\boldsymbol{\tau}_{1:t-1}$ by constraining the combined parameter $\boldsymbol{\sigma} \otimes \mathcal{M}_i + \boldsymbol{\tau}_i$ to be close to $\boldsymbol{\theta}_i^*$. The learning objective for the current task $t$ is then described as follows:

$$\underset{\boldsymbol{\sigma}, \boldsymbol{\tau}_{1:t}, \mathbf{v}_{1:t}}{\text{minimize}} \mathcal{L}\left(\{\boldsymbol{\sigma} \otimes \mathcal{M}_t + \boldsymbol{\tau}_t\}; \mathcal{D}_t\right) + \lambda_1 \sum_{i=1}^{t} \|\boldsymbol{\tau}_i\|_1 + \lambda_2 \sum_{i=1}^{t-1} \|\boldsymbol{\theta}_i^* - (\boldsymbol{\sigma} \otimes \mathcal{M}_i + \boldsymbol{\tau}_i)\|_2^2. \tag{2}$$

Compared to (1), the task-adaptive parameters of previous tasks now can be retroactively updated to minimize the parameter-level drift. This formulation also constrains the update of the task-shared parameters to consider order-robustness.

---

**Algorithm 1** Continual learning with Additive Parameter Decomposition

---

**input** Dataset $\mathcal{D}_{1:T}$ and hyperparameter $\lambda, m, s, K = k$
**output** $\boldsymbol{\sigma}^{(T)}$, $\mathbf{v}_{1:T}$, $\widetilde{\boldsymbol{\sigma}}_{1:K}$, and $\boldsymbol{\tau}_{1:T}$
1: Let $\boldsymbol{\sigma}^{(1)} = \boldsymbol{\theta}_1$, and optimize for the task 1
2: **for** $t = 2, ..., T$ **do**
3:     **for** $i = 1, ..., t-1$ **do**
4:        Restore $\boldsymbol{\theta}_i^* = \boldsymbol{\sigma}^{(t-1)} \otimes \mathcal{M}_i^{(t-1)} + \widetilde{\boldsymbol{\tau}}_i^{(t-1)}$
5:     **end for**
6:     Minimize (3) to update $\boldsymbol{\sigma}$ and $\{\boldsymbol{\tau}_i, \mathbf{v}_i\}_{i=1}^t$
7:     **if** $t \mod s = 0$ **then**
8:        Initialize $k$ new random centroids, $\{\boldsymbol{\mu}_g\}_{g=K-k+1}^K$
9:        Group all tasks into $K$ disjoint sets, $\{\mathcal{G}_g\}_{g=1}^K$
10:       **for** $g = 1, ..., K$ **do**
11:          Decompose $\{\widetilde{\boldsymbol{\tau}}_i\}_{i \in \mathcal{G}_g}$ into $\widetilde{\boldsymbol{\sigma}}_g$ and $\{\boldsymbol{\tau}_i\}_{i \in \mathcal{G}_g}$
12:       **end for**
13:       Delete old $\widetilde{\boldsymbol{\sigma}}$ and $K = K + k$
14:     **end if**
15: **end for**

---

**Hierarchical Knowledge Consolidation**    The objective function in (2) does not directly consider local sharing among the tasks, and thus it will inevitably result in the redundancy of information in the task-adaptive parameters. To further minimize the capacity increase, we perform a process called *hierarchical knowledge consolidation* to group relevant task-adaptive parameters into task-shared parameters (See Figure 2). We first group all tasks into $K$ disjoint sets $\{\mathcal{G}_g\}_{g=1}^K$ using $K$-means clustering on $\{\boldsymbol{\tau}_i\}_{i=1}^t$, then decompose the task-adaptive parameters in the same group into locally-shared parameters $\widetilde{\boldsymbol{\sigma}}_g$ and task-adaptive parameters $\{\boldsymbol{\tau}_i\}_{i \in \mathcal{G}_g}$ (with higher sparsity) by simply computing the amount of value discrepancy in each parameter as follows:

- If $\max \{\boldsymbol{\tau}_{i,j}\}_{i \in \mathcal{G}_g} - \min \{\boldsymbol{\tau}_{i,j}\}_{i \in \mathcal{G}_g} \leq \beta$, then $\{\boldsymbol{\tau}_{i,j}\}_{i \in \mathcal{G}_g} = 0$ and $\widetilde{\boldsymbol{\sigma}}_{g,j} = \boldsymbol{\mu}_{g,j}$
- Else, $\widetilde{\boldsymbol{\sigma}}_{g,j} = 0$,

where $\boldsymbol{\tau}_{i,j}$ denotes the $j$th element of the $i$th task-adaptive parameter matrix, and $\boldsymbol{\mu}_g$ is the cluster center of group $\mathcal{G}_g$. We update the locally-shared parameters $\widetilde{\boldsymbol{\sigma}}_g$ after the arrival of every $s$ tasks for efficiency, by performing $K$-means clustering while initializing the cluster centers with the previous locally-shared parameters $\widetilde{\boldsymbol{\sigma}}_g$ for each group. At the same time, we increase the number of centroids to $K + k$ to account for the increase in the variance among the tasks.

Our final objective function is then given as follows:

$$\underset{\boldsymbol{\sigma}, \boldsymbol{\tau}_{1:t}, \mathbf{v}_{1:t}}{\text{minimize}} \; \mathcal{L}\left(\{\boldsymbol{\sigma} \otimes \mathcal{M}_t + \boldsymbol{\tau}_t\}; \mathcal{D}_t\right) + \lambda_1 \sum_{i=1}^t \|\boldsymbol{\tau}_i\|_1 + \lambda_2 \sum_{i=1}^{t-1} \|\boldsymbol{\theta}_i^* - (\boldsymbol{\sigma} \otimes \mathcal{M}_i + \widetilde{\boldsymbol{\tau}}_i)\|_2^2, \quad (3)$$
$$\text{where } \widetilde{\boldsymbol{\tau}}_i = \boldsymbol{\tau}_i + \widetilde{\boldsymbol{\sigma}}_g \text{ for } i \in \mathcal{G}_g.$$

Algorithm 1 describes the training of our APD model.

**Selective task forgetting**    In practical scenarios, some of earlier learned tasks may become irrelevant as we continually train the model. For example, when we are training a product identification model, recognition of discontinued products will be unnecessary. In such situations, we may want to forget the earlier tasks in order to secure network capacity for later task learning. Unfortunately, existing continual learning methods cannot effectively handle this problem, since the removal of some features or parameters will also negatively affect the remaining tasks as their parameters are entangled. Yet, with APDs, forgetting of a task $t$ can be done by dropping out the task adaptive parameters $\boldsymbol{\tau}_t$. Trivially, this will have absolutely no effect on the task-adaptive parameters of the remaining tasks.

## 4 EXPERIMENT

We now validate APD-Net on multiple datasets against state-of-the-art continual learning methods.

### 4.1 DATASETS

**1) CIFAR-100 Split** (Krizhevsky & Hinton, 2009) consists of images from 100 generic object classes. We split the classes into 10 group, and consider 10-way multi-class classification in each group as a single task. We use 5 random training/validation/test splits of $4,000/1,000/1,000$ samples.

**2) CIFAR-100 Superclass** consists of images from 20 superclasses of the CIFAR-100 dataset, where each superclass consists of 5 different but semantically related classes. For each task, we use 5 random training/validation/test splits of $2,000/500/500$ samples.

**3) Omniglot-rotation** (Lake et al., 2015) contains OCR images of $1,200$ characters (we only use the training set) from various writing systems for training, where each class has 80 images, including 0, 90, 180, and 270 degree rotations of the original images. We use this dataset for large-scale continual learning experiments, by considering the classification of 12 classes as a single task, obtaining 100 tasks in total. For each class, we use 5 random training/test splits of $60/20$ samples.

We use a modified version of LeNet-5 (LeCun et al., 1998) and VGG16 network (Simonyan & Zisserman, 2015) with batch normalization as base networks. For experiments on more datasets, and detailed descriptions of the architecture and task order sequences, please see the **supplementary file**.

### 4.2 BASELINES AND OUR MODELS

**1) L2-Transfer.** Deep neural networks trained with the $L2$-transfer regularizer $\lambda\|\boldsymbol{\theta}_t - \boldsymbol{\theta}_{t-1}\|_F^2$ when training for task $t$. **2) EWC.** Deep neural networks regularized with Elastic Weight Consolidation (Kirkpatrick et al., 2017). **3) P&C.** Deep neural networks with two-step training: Progress, and Compresss (Schwarz et al., 2018). **4) PGN.** Progressive Neural Networks (Rusu et al., 2016) which constantly increase the network size by $k$ neurons with each task. **5) DEN.** Dynamically Expandable Networks (Yoon et al., 2018) that selectively retrain and dynamically expand the network size by introducing new units and duplicating neurons with semantic drift. **6) RCL.** Reinforced Continual Learning proposed in (Xu & Zhu, 2018) which adaptively expands units at each layer using reinforcement learning. **7) APD-Fixed.** APD-Net without the retroactive update of the previous task-adaptive parameters (Eq. (1)). **8) APD(**1**).** Additive Parameter Decomposition Networks with depth 1, whose parameter is decomposed into task-shared and task-adaptive parameters. **10) APD(**2**).** APD-Net with depth 2, that also has locally shared parameters from hierarchical knowledge consolidation.

### 4.3 QUANTITATIVE EVALUATION

**Task-average performance**  We first validate the final task-average performance after the completion of continual learning. To perform fair evaluation of performance that is not order-dependent, we report the performance on three random trials over 5 different task sequences over all experiments. Table 1 shows that APD-Nets outperform all baselines by large margins in accuracy. We attribute this performance gain to two features. First, an APD-Net uses neuron(filter)-wise masking on the shared parameters, which allows it to focus only on parts that are relevant to the task at the current training stage. Secondly, an APD-Net updates the previous task-adaptive parameters to reflect the changes made to the shared parameters, to perform retroactive knowledge transfer. APD-Fixed, without these retroactive updates, performs slightly worse. APD(2) outperforms APD(1) since it further allows local knowledge transfer with hierarchically shared parameters. Moreover, when compared with expansion based baselines, our methods yield considerably higher accuracy with lower capacity (Figure 3). This efficiency comes from the task-adaptive learning performing only residual learning for each task with minimal capacity increase, while maximally utilizing the task-shared parameters.

We further validate the efficiency of our methods in terms of training time. Existing approaches with network expansion are slow to train. DEN should be trained with multiple steps, namely selective retraining, dynamic network expansion and split/duplication, each of which requires retraining of the network. RCL is trained with reinforcement learning, which is inherently slow since the agent should determine exactly how many neurons to add at each layer in a discrete space. PGN trains much faster, but the model increases the fixed number of neurons at each layer when a new task arrives, resulting in overly large networks. On the contrary, APD-Net, although it requires updates to the previous task-adaptive parameters, can be trained in a single training step. Figure 3 shows that both APD(1) and APD(2) have training time comparable to the base model, with only a marginal increase.

Table 1: Experiment results on CIFAR-100 Split and CIFAR-100 Superclass datasets. The results are the mean accuracies over 3 runs of experiments with random splits, performed with 5 different task order sequences. STL is the single-task learning model that trains a separate network for each task independently. Standard deviations for accuracy are given in Table A.3 in the Appendix.

| Methods | CIFAR-100 Split | | | | CIFAR-100 Superclass | | | |
|---|---|---|---|---|---|---|---|---|
| | Capacity | Accuracy | AOPD | MOPD | Capacity | Accuracy | AOPD | MOPD |
| STL | 1,000% | 63.75% | 0.98% | 2.23% | 2,000% | 61.00% | 2.31% | 3.33% |
| L2T | 100% | 48.73% | 8.62% | 17.77% | 100% | 41.40% | 8.59% | 20.08% |
| EWC | 100% | 53.72% | 7.06% | 15.37% | 100% | 47.78% | 9.83% | 16.87% |
| P&C | 100% | 53.54% | 6.59% | 11.80% | 100% | 48.42% | 9.05% | 20.93% |
| PGN | 171% | 54.90% | 8.08% | 14.63% | 271% | 50.76% | 8.69% | 16.80% |
| DEN | 181% | 57.38% | 8.33% | 13.67% | 191% | 51.10% | 5.35% | 10.33% |
| RCL | 181% | 55.26% | 5.90% | 11.50% | 184% | 51.99% | 4.98% | 14.13% |
| APD-Fixed | 132% | 59.32% | 2.43% | 4.03% | 128% | 55.75% | 3.16% | 6.80% |
| APD(1) | 134% | **59.93%** | **2.12%** | **3.43%** | 133% | **56.76%** | **3.02%** | **6.20%** |
| APD(2) | 135% | **60.74%** | **1.79%** | **3.43%** | 130% | **56.81%** | **2.85%** | **5.73%** |

(a) CIFAR-100 Split (T=10)  (b) CIFAR-100 Superclass (T=20)

Figure 3: Accuracy over efficiency of expansion-based continual learning methods and our methods. We report performance over capacity and performance over training time on both datasets.

**Order fairness in continual learning**   We now evaluate the order-robustness of our model in comparison to the existing approaches. We first define an evaluation metric for order-sensitivity for each task $t$, which we name as *Order-normalized Performance Disparity (OPD)*, as the disparity between its performance on $R$ random task sequences:

$$OPD_t = \max(\overline{P}_t^1, ..., \overline{P}_t^R) - \min(\overline{P}_t^1, ..., \overline{P}_t^R) \tag{4}$$

where $\overline{P}_t^r$ denotes the performance of task $t$ to the task sequence $r$. Then we define the *Maximum OPD* as $MOPD = \max(OPD_1, ..., OPD_t)$, and the *Average OPD* as $AOPD = \frac{1}{T}\sum_{t=1}^{T} OPD_t$, to evaluate order-robustness on the entire task set. A model that is sensitive to the task sequence order will have high MOPD and AOPD, and an order-robust model will have low values for both metrics.

In table 1, we show the experimental results on order-robustness for all models, obtained on 5 random sequences. We observe that expansion-based continual learning methods are more order-robust than fixed-capacity methods, owing to their ability to introduce task-specific units, but they still suffer from a large degree of performance disparity due to asymmetric direction of knowledge transfer from earlier tasks to later ones. On the other hand, APD-Nets obtain significantly lower MOPD and AOPD compared to baseline models that have high performance disparity between task sequences given in different orders. APD(1) and APD(2) are more order-robust than APD-Fixed, which suggests the effectiveness of the retroactive updates of $\tau_{1:t-1}$. Figure 4 further shows how the per-task performance of each model changes to task sequences of three different orders. We observe that our models show the least disparity in performance to the order of the task sequence.

**Preventing catastrophic forgetting**   We show the effectiveness of APD on its prevention of catastrophic forgetting by examining how the model performance on earlier tasks change as new tasks arrive. Figure 5, (a)-(c) show the results on task $1, 6, 11$ from CIFAR-100 Superclass, which has 20 tasks in total. APD-Nets do not show any sign of catastrophic forgetting, although their performances marginally change with the arrival of each task. In fact, APD(2) even improves on task 6 (by $0.40\%p$) as it learns on later tasks, which is possible both due to the update of the shared parameters and the retroactive update of the task-adaptive parameters for earlier tasks, which leads to better solutions.

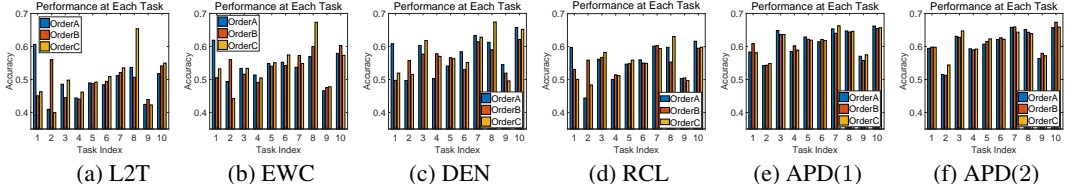

Figure 4: Performance disparity of continual learning baselines and our models on CIFAR-100 Split. Plots show per-task accuracy for 3 task sequences of different order. Performance disparity of all methods for 5 task sequences of different order are given in Figure A.8 in the Appendix.

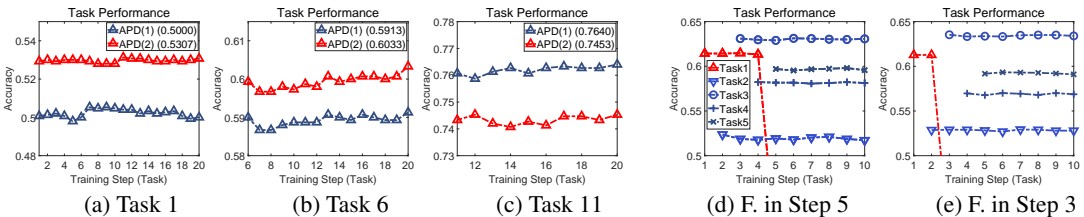

Figure 5: **(a)-(c) Catastrophic Forgetting** on CIFAR-100 Superclass: Performance of our models on the $1^{st}$, $6^{th}$, and $11^{th}$ task during continual learning. **(d)-(e) Task Forgetting** on CIFAR-100 Split: Per-task Performance of APD(1) ($T_{1:5}$) when $1^{st}$ task is dropped during continual learning.

**Selective task forgetting** To show that APD-Net can perform selective task forgetting without any harm on the performance of non-target tasks, in Figure 5, (d)-(e), we report the performance change in Task 1-5 when removing parameters for Task 3 and 5. As shown, there is no performance degeneration on non-target tasks, which is expected since dropping out a task-adaptive parameter for a specific task will not affect the task-adaptive parameters for the remaining tasks. This ability to selectively forget is another important advantage of our model that makes it practical in lifelong learning scenarios.

**Scalability to large number of tasks** We further validate the scalability of our model with large-scale continual learning experiments on the Omniglot-Rotation dataset, which has 100 tasks. Regardless of random rotations, tasks could share specific features such as circles, curves, and straight lines. Gidaris et al. (2018) showed that we can learn generic representations even with rotated images, where they proposed a popular self-supervised learning technique where they train the model to predict the rotation angle of randomly rotated images. We do not compare against DEN or RCL for this experiment since they are impractically slow to train. Figure 6 (Left) shows the results of this experiment. For PGN, we restrict the maximum number of links to the adapter to 3 in order to avoid it from establishing exponentially many connections. We observe that continual learning models achieve significantly lower performance and high OPDs compared to single task learning. On the contrary, our model outperforms them by large amount, obtaining performance that is almost equal to STL which uses 100 times more network parameters. To show that our model scales well, we plot the number of parameters for our models as a function of the number of tasks in Figure 6 (Right). The plot shows that our APD-Net scales well, showing logarithmic growth in network capacity (the number of parameters), while PGN shows linear growth. This result suggests that our model is highly efficient especially in large-scale continual learning scenarios.

**Continual learning with heterogenerous datasets** We further consider a more challenging continual learning scenario where we train on a series of heterogeneous datasets. For this experiment, we use CIFAR-10 (Krizhevsky & Hinton, 2009), CIFAR100, and the Street View House Numbers (SVHN) (Netzer et al., 2011) dataset, in two different task arrival sequences (SVHN→CIFAR-10→CIFAR-100, CIFAR-100→CIFAR-10→SVHN). We use VGG-16 as the base network, and compare against an additional baseline, Piggyback (Mallya et al., 2018), which handles a newly arrived task by learning a task-specific binary mask on a network pretrained on ImageNet; since we cannot assume the availability of such large-scale datasets for pretraining in a general setting, we pretrain it on the inital task. Table 2 shows the results, which show that existing models obtain

| Models | Capacity | Accuracy | AOPD | MOPD |
|---|---|---|---|---|
| STL | 10,000% | 82.13% (0.08) | 2.79% | 5.70% |
| L2T | 100% | 63.46% (1.58) | 13.35% | 24.43% |
|  | 1,599% | 64.65% (1.76) | 11.35% | 27.23% |
| EWC | 100% | 67.48% (1.39) | 14.92% | 32.93% |
|  | 1,599% | 68.66% (1.92) | 15.19% | 40.43% |
| PGN | 1,045% | 73.65% (0.27) | 6.79% | 19.27% |
|  | 1,543% | 79.35% (0.12) | 4.52% | 10.37% |
| APD(2) | **649%** | **81.20% (0.62)** | **4.09%** | **9.44%** |
|  | **943%** | **81.60% (0.53)** | **3.78%** | **8.19%** |

Figure 6: **Left:** Performance comparison with several benchmarks on Omniglot-rotation (standard deviation into parenthesis). **Right:** The number of the parameters which is obtained during course of training on Omniglot-rotation.

Table 2: Accuracy comparison on diverse datasets according to two opposite task order (arrows). The results are the mean accuracies over 3 runs of experiments. VGG16 with batch normalization is used for a base network.

| Task Order | STL | L2T | | Piggyback | | PGN | | APD(1) | |
|---|---|---|---|---|---|---|---|---|---|
|  | None | ↓ | ↑ | ↓ | ↑ | ↓ | ↑ | ↓ | ↑ |
| SVHN | 96.8% | 10.7% | 88.4% | **96.8%** | 96.4% | **96.8%** | 96.2% | **96.8%** | **96.8%** |
| CIFAR10 | 91.3% | 41.4% | 35.8% | 83.6% | 90.8% | 85.8% | 87.7% | **90.1%** | **91.0%** |
| CIFAR100 | 67.2% | 29.6% | 12.2% | 41.2% | **67.2%** | 41.6% | **67.2%** | 61.1% | **67.2%** |
| Average | 85.1% | 27.2% | 45.5% | 73.9% | 84.8% | 74.7% | 83.7% | **83.0%** | **85.0%** |
| Model Size | 171MB | 57 MB | 57 MB | 59 MB | 59 MB | 64 MB | 64 MB | 63 MB | 65 MB |

suboptimal performance in this setting and are order-sensitive. While Piggyback and PGN are immue to catastrophic forgetting since they freeze the binary masks and hidden units trained on previous tasks, they still suffer from performance degeneration, since their performances largely depends upon the pretrained network and the similarity of the later tasks to earlier ones. On the contrary, APD obtains performance close to STL without much increase to the model size, and is also order-robust.

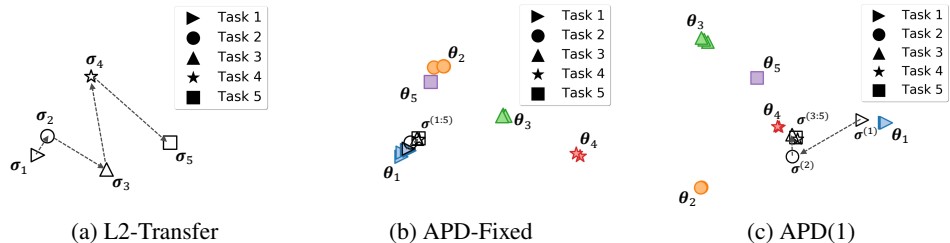

| (a) L2-Transfer | (b) APD-Fixed | (c) APD(1) |
|---|---|---|

Figure 7: **Visualizations of the model paramters during continual learning.** The colored markers denote the parameters for each task $i$, and the empty markers with black outlines denote the task-shared parameters. Dashed arrows indicate the drift in the parameter space as the model trains on a sequence of tasks.

## 4.4 QUALITATIVE ANALYSIS

As a further qualitative analysis of the effect of APD, we visualize the parameters using our method and baselines by projecting them onto a 2D space (Figure 7). For this experiment, we use a modified MNIST-split dataset whose images are cropped in the center by $8 \times 8$ pixels, and create $5$ tasks, where each task is the binary classification between two classes. As for the base network, we use a 2-layer multi-layer perceptron with $10$ units at each layer. Then we use Principle Component Analysis (PCA) to reduce the dimensionality of the parameters to two. We visualize the 2D projections of both the task-shared and task-adaptive parameters for each step of continual learning. For example, for task 3, we plot three green markers which visualize teh parameters when training on task 4 and 5. For the last task (Task 5), we only have a single marker since this is the last task.

We observe that the model parameters using $L2$-Transfer drift away in a new direction, as it trains on a sequence of tasks, which brings in catastrophic forgetting. APD-Fixed (Figure 7(b)) largely alleviates the semantic drift, as the update on later tasks only affects the task-shared parts while the task-adaptive parameters are kept intact. However, the update to the task-shared parameters could result in small drift in the combined task-specific parameters. On the other hand, APD-Net with retroactive update of task-adaptive parameters successfully prevents the drift in the task-specific parameters (Figure 7(c)) .

## 5  CONCLUSION

We proposed a novel continual learning model with Additive Parameter Decomposition, where the task-shared parameters capture knowledge generic across tasks and the task-adaptive parameters capture incremental differences over them to capture task-specific idiosyncrasies. This knowledge decomposition naturally solves the catastrophic forgetting problem since the task-adaptive parameters for earlier tasks will remain intact, and is significantly more efficient compared to expansion-based approaches, since the task-adaptive parameters are additive and do not increase the number of neurons or filters. Moreover, we also introduce and tackle a novel problem we refer to as *task order sensitivity*, where the performance for each task varies sensitively to the order of task arrival sequence; with our model, the shared parameters will stay relatively static regardless of the task order, and retroactive updates of the task-adaptive parameters prevent them from semantic drift. With extensive experimental validation, we showed that our model obtains impressive accuracy gains over the existing continual learning approaches, while being memory- and computation-efficient, scalable to large number of tasks, and order-robust. We hope that our paper initiates new research directions for continual learning on the relatively unexplored problems of scalability, task-order sensitivity, and selective task forgetting.

**Acknowledgements**  This work was supported by Samsung Advanced Institute of Technology, Samsung Research Funding Center of Samsung Electronics (No. SRFC-IT1502-51), the Engineering Research Center Program through the National Research Foundation of Korea (NRF) funded by the Korean Government MSIT (NRF-2018R1A5A1059921), the National Research Foundation of Korea (No. NRF-2016M3C4A7952634, Development of Machine Learning Framework for Peta Flops Scale), Institute for Information  communications Technology Promotion (IITP) grant funded by the Korea government (MSIT) (No.2016-0-00563, Research on Adaptive Machine Learning Technology Development for Intelligent Autonomous Digital Companion), and Institute of Information  communications Technology Planning  Evaluation (IITP) grant funded by the Korea government (MSIT) (No.2019-0-00075, Artificial Intelligence Graduate School Program (KAIST)).

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

## A  APPENDIX

We introduce detailed experiment settings for our Additive Parameter Decomposition (APD). Also, we provide experimental results including additional quantitative analysis and ablation study for our model.

### A.1  EXPERIMENT SETTINGS

In this section, we describe experimental details for our models. We used exponential learning rate decay at each epoch and all models are applied on weight decay with $\lambda = 1e^{-4}$. All hyperparameters are determined from a validation set. All experiments are performed without data preprocessing techniques. For MNIST-Variation, we used two-layered feedforward networks with 312, 128 neurons. Training epochs are 50 for all baselines and APDs. $\lambda_1 = [2e^{-4},\ 1e^{-4}]$ on APD.

For CIFAR-100 Split and CIFAR-100 Superclass, we used LeNet with 20-50-800-500 neurons. Training epochs are 20 for all models. $\lambda_1 = [6e^{-4},\ 4e^{-4}]$. We equally set $\lambda_2 = 100$, also $K$=2 per 5 tasks, and $\beta$=$1e^{-2}$ for hierarchical knowledge consolidation on MNIST-Variation, CIFAR-100 Split, and CIFAR-100 Superclass.

For Omniglot, we used LeNet with 10-20-500-300 neurons as default. And to show the performance EWC with larger network capacity, we used LeNet with 64-128-2500-1500 neurons. Training epochs are 100 for all models, and $\lambda_1 = [4e^{-4},\ 2e^{-4}]$, and $\lambda_2 = 100$, and 1K for APD. We set $K$=3 per 10 tasks, and $\beta$=$1e^{-4}$ for hierarchical knowledge consolidation. Note that we use an additional technique which updates only largely changed $\boldsymbol{\theta}_i$ where $i < t$. It bypasses the retroactive parameter update for the tasks which is nearly relevant to learn the current task $t$. This selective update rule helps the model skip these meaningless update procedure and we can train our model much faster on large-scale continual learning.

To estimate order robustness, we used 5 different orders on all experiments. For the case of MNIST-Variation and CIFAR-100 Split, we select random generated orders as follows:

- orderA: $[0, 1, 2, 3, 4, 5, 6, 7, 8, 9]$
- orderB: $[1, 7, 4, 5, 2, 0, 8, 6, 9, 3]$
- orderC: $[7, 0, 5, 1, 8, 4, 3, 6, 2, 9]$
- orderD: $[5, 8, 2, 9, 0, 4, 3, 7, 6, 1]$
- orderE: $[2, 9, 5, 4, 8, 0, 6, 1, 3, 7]$

For CIFAR-100 Superclass, we select random generated orders as follows:

- orderA: $[0, 1, 2, 3, 4, 5, 6, 7, 8, 9, 10, 11, 12, 13, 14, 15, 16, 17, 18, 19]$
- orderB: $[15, 12, 5, 9, 7, 16, 18, 17, 1, 0, 3, 8, 11, 14, 10, 6, 2, 4, 13, 19]$
- orderC: $[17, 1, 19, 18, 12, 7, 6, 0, 11, 15, 10, 5, 13, 3, 9, 16, 4, 14, 2, 8]$
- orderD: $[11, 9, 6, 5, 12, 4, 0, 10, 13, 7, 14, 3, 15, 16, 8, 1, 2, 19, 18, 17]$
- orderE: $[6, 14, 0, 11, 12, 17, 13, 4, 9, 1, 7, 19, 8, 10, 3, 15, 18, 5, 2, 16]$

For Omniglot dataset, we omit the sequence of random generated orders for readability.

Table A.3: Ablation study results on APD(1) with average of five different orders depicted in **A.1**. We show a validity of APD as comparing with several architectural variants. All experiments performed on CIFAR-100 split dataset.

| Models | Capacity | Accuracy | AOPD | MOPD |
|---|---|---|---|---|
| STL | 1,000% | 63.75% | 0.98% | 2.23% |
| APD(1) | 170% | **61.30%** | **1.57%** | **2.77%** |
| w/o Sparsity | 1,084% | **63.47%** | 3.20% | 5.40% |
| w/o Adaptive Mask | 168% | 59.09% | 1.83% | 3.47% |
| Fixed $\sigma$ | 167% | 58.55% | 2.31% | 3.53% |

## A.2 ARCHITECTURAL CHOICES FOR ADDITIVE PARAMETER DECOMPOSITION

We also evaluate various ablation experiments on Additive Parameter Decomposition. First of all, we build the dense APD without sparsity inducing constraints for task-adaptive parameters while maintaining the essential architecture, depicted as **w/o Sparsity**. It significantly outperforms APD in terms of accuracy but impractical since it requires huge capacity. We also measure the performance of APD without adaptive masking variables to observe how much performance is degraded when the flexibility of APD for newly arriving tasks is limited, which is referred to as **w/o Adaptive Masking** in the table. Naturally, it underperforms with respect to both accuracy and OPDs. Freezing $\sigma$ after training the first task, referred to as **Fixed** $\sigma$ in the table, is designed to observe the performance when the task-shared knowledge is not properly captured by $\sigma$. Interestingly, this shows much lower performance than other variants, suggesting that it is extremely crucial to properly learn the task-shared knowledge during continual learning.

Table A.4: Comparison with GEM-variants on Permuted-MNIST dataset. We followed all experimental settings from A-GEM (Chaudhry et al., 2019). We report the performance on single epoch training for 17 random permuted MNIST except 3 cross-validation tasks from 20 total tasks, mini-batch is 10 and size of episodic memory in GEMs is 256. We refered the experimental results for GEM variants from Chaudhry et al. (2019).

| Methods | Network Capacity (%) | Accuracy | Average Forgetting | Worst-case Forgetting |
|---------|---------------------|----------|-------------------|----------------------|
| STL | 1,700% | 0.9533 | 0.00 | 0.00 |
| GEM | 100% | 0.8950 | 0.060 | 0.100 |
| S-GEM | 100% | 0.8820 | 0.080 | - |
| A-GEM | 100% | 0.8910 | 0.060 | 0.130 |
| APD(1) | 103% | **0.9067** | **0.020** | **0.051** |
| APD(1) | 115% | **0.9283** | **0.018** | **0.047** |

Table A.5: Comparison with HAT (Serrà et al., 2018) on sequence of 8 heterogeneous dataset. We follow all experimental settings from HAT and reproduce the performance of HAT directly from the author's code. We perform the experiments with 5 different (randomly generated) task order sequences. We use forgetting measure as Average Forgetting and Worst-case Forgetting from (Chaudhry et al., 2019).

| Methods | Network Capacity | Accuracy | Average F. | Worst-case F. | AOPD | MOPD |
|---------|-----------------|----------|-----------|---------------|------|------|
| HAT | 100% | 0.8036 (0.012) | 0.0014 | 0.0050 | 0.0795 | 0.2315 |
| HAT-Large | 182% | 0.8183 (0.011) | 0.0013 | 0.0057 | 0.0678 | 0.1727 |
| APD-Fixed | 181% | **0.8242 (0.005)** | **0.0003** | **0.0006** | **0.0209** | **0.0440** |

## A.3 COMPARISON WITH OTHER CONTINUAL LEARNING METHODS

We additionally compare our APD with GEM-based approaches (Lopez-Paz & Ranzato, 2017; Chaudhry et al., 2019). As for the backbone networks, we use a two-layer perceptron with 256 neurons at each layer. The results in Table A.4 show that GEM-variants obtain reasonable performance with a marginal forgetting since the models store data instances of previous tasks in the episodic memory, and use them to compute gradients for training on later tasks. Note that we do not count the size of episodic memory to the network capacity.

Furthermore, we compare APD-Net against HAT (Serrà et al., 2018) on a sequence of 8 heterogeneous dataset including CIFAR-10, CIFAR-100, FaceScrub (Ng & Winkler, 2014), MNIST (LeCun et al., 1998), NotMNIST (Bulatov, 2011), FashionMNIST (Xiao et al., 2017), SVHN, and TrafficSign (Stallkamp et al., 2011). We used a modified version of AlexNet (Krizhevsky et al., 2012) as the backbone networks and reproduce the performance of HAT directly from the author's code. Table A.5 shows that APD-Fixed largely outperforms HAT.

Both GEM variants and HAT are strong continual learning approaches, but cannot expand the network capacity and/or performs unidirectional knowledge transfer thus suffers from the capacity limitation and order-sensitivity. On the other hand, our APD adaptively increases the network capacity by introducing task-adaptive parameters which learns task-specific features not captured in the task-shared parameters. Therefore, APD can learn richer representations compared to fixed-capacity continual learning approaches. APD also exhibit several unique properties, such as task-order robustness and trivial task forgetting.

Table A.6: Full experiment results on CIFAR-100 Split and CIFAR-100 Superclass datasets. The results are the mean accuracies over 3 runs of experiments with random splits, preformed with 5 different task order sequences (standard deviation into parenthesis).

| Methods | Capacity | Accuracy | AOPD | MOPD |
|---|---|---|---|---|
| | | **CIFAR-100 Split** | | |
| STL | 1,000% | 63.75% (0.14) | 0.98% | 2.23% |
| L2T | 100% | 48.73% (0.66) | 8.62% | 17.77% |
| EWC | 100% | 53.72% (0.56) | 7.06% | 15.37% |
| P&C | 100% | 53.54% (1.70) | 6.59% | 11.80% |
| PGN | 171% | 54.90% (0.92) | 8.08% | 14.63% |
| DEN | 181% | 57.38% (0.56) | 8.33% | 13.67% |
| RCL | 181% | 55.26% (0.13) | 5.90% | 11.50% |
| APD-Fixed | 132% | 59.32% (0.44) | 2.43% | 4.03% |
| | 175% | **61.02% (0.31)** | 2.26% | **2.87%** |
| APD(1) | 134% | 59.93% (0.41) | 2.12% | 3.43% |
| | 170% | **61.30% (0.37)** | **1.57%** | **2.77%** |
| APD(2) | 135% | **60.74% (0.21)** | **1.79%** | 3.43% |
| | 153% | **61.18% (0.20)** | **1.86%** | **3.13%** |
| | | **CIFAR-100 Superclass** | | |
| STL | 2,000% | 61.00% (0.20) | 2.31% | 3.33% |
| L2T | 100% | 41.40% (0.99) | 8.59% | 20.08% |
| EWC | 100% | 47.78% (0.74) | 9.83% | 16.87% |
| P&C | 100% | 48.42% (1.39) | 9.05% | 20.93% |
| PGN | 271% | 50.76% (0.39) | 8.69% | 16.80% |
| DEN | 191% | 51.10% (0.77) | 5.35% | 10.33% |
| RCL | 184% | 51.99% (0.25) | 4.98% | 14.13% |
| APD-Fixed | 128% | 55.75% (1.01) | 3.16% | 6.80% |
| | 191% | **57.98% (0.65)** | 2.58% | **4.53%** |
| APD(1) | 133% | 56.76% (0.27) | 3.02% | 6.20% |
| | 191% | **58.37% (0.22)** | 2.64% | **5.47%** |
| APD(2) | 130% | 56.81% (0.33) | 2.85% | 5.73% |
| | 182% | **58.53% (0.31)** | 2.75% | **5.67%** |

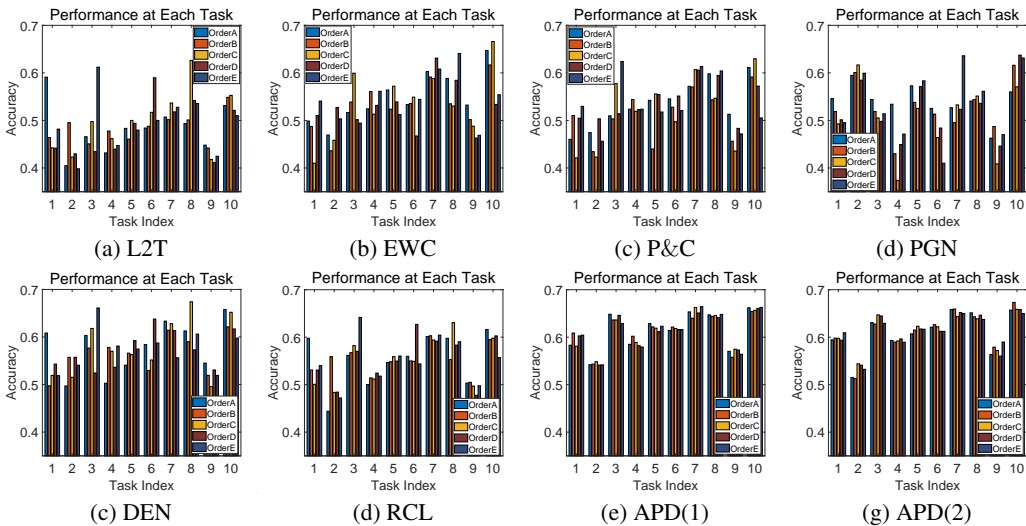

(a) L2T    (b) EWC    (c) P&C    (d) PGN

(c) DEN    (d) RCL    (e) APD(1)    (g) APD(2)

Figure A.8: Per-task accuracy for each task sequence of continual learning baselines and our models on CIFAR-100 Split, on 5 task sequences of different order. Large amount of disparity among task performance of different orders implies that the model is task-order sensitive, that is less confident in terms of fairness in continual learning.

