# OpenReview forum: "Scalable and Order-robust Continual Learning with Additive Parameter Decomposition"
_ICLR.cc/2020/Conference — Accept (Poster)_

### Official Review · AnonReviewer1 · 2019-10-22
**Official Blind Review #1**

**Rating:** 6

**Review:**

The paper proposes a training framework that:
(i) can efficiently handle catastrophic forgetting in a large number of tasks
(ii) is robust to the ordering of the tasks

The high-level idea is to decompose learning parameters into two sets - one set that depends on the task and one set that is task agnostic. Hierarchal clustering is used to improve the efficiency of the training process by considering the decomposition of parameters at multiple levels (and not just 2). The paper shows improvements in terms of accuracy, stability, and order-robustness and provides ablation results (for various modifications of their proposed model) and considers a setup with around 100 tasks. The paper/idea is quite interesting and the results seem promising:

* The results in figure 5: (d) and (e) are very interesting. It seems as if all the "previous" knowledge is being contained in the task-specific parameters. In general, I like the idea of being able to "forget" all the previous knowledge. I want to clarify one thing here: My understanding is that after training the model on tasks 1 to 5, the weights corresponding to task 1 are dropped (that is just the task-specific parameters tau and not the mask). Then before even one gradient update is applied, the model is re-evaluated on task 1. Is that correct?

* Figure 7 seems to suggest that the drift is reduced by the proposed approach. What does the presence of multiple markers (of the same color) mean? For example, there are two green triangles in (b).

====================

But, many things should be clarified for understanding the paper properly and for making a fair assessment of the claims made in the paper. I would be happy to update my score based on the authors' response to the following:

* My biggest concern is the choice of baselines. The paper (rightly)  highlights that their work improves over many existing works as they provide a mechanism to "retroactively update task-adaptive parameters" for the previous task. But none of their baselines have this mechanism built-in. So while there is a clear advantage with the proposed approach, the comparison is unfair and the baselines should have considered approaches like GEM [0[ and A-GEM[1] while also provide a kind-of retrospective mechanism to correct the weights corresponding to the subsequent tasks. Without such a comparison, it is difficult to comment on the benefits of the approach.

* On page 2, paragraph 3, the paper mentions that "APD does not increase the intrinsic network complexity as existing expansion-based approaches do". This claim seems to be loose since a new mask is learned for each task and needs to be persisted for all the subsequent tasks. So there is an "expansion"-like step involved. The authors should clarify this detail in the context of being memory efficient.

* The authors propose a simplistic regularization approach (L2) to ensure that the shared parameters do not share too much across tasks. While it is good that a simple approach works so well, it would help if the authors discussed what do they think the reason is. EWC [2] and the authors' results indicate that regularization in the parameter space does not work as well as regularization in the function space. Thus the L2 regularization approach is not likely to work well.

* Some parts of the paper needs to be reworded or clarified more. For example, since a per-task mask is being learned, there are two sets of weights being learned per task (the mask and the task-specific parameters). So there are more parameters to learn and not just the sparse task-specific parameters.

* As I understand, the paper uses "order robustness"  to mean avoiding "concept drift" (or "catastrophic forgetting"). I might be missing something (in plain sight) but when I think of "order robustness", the order of tasks should not matter. This is somewhat different than avoiding concept drift.

* Is the hierarchical clustering being done for each neuron (of the given model)? If yes, how does this approach scale to large neural nets? In general, how does the cost of doing the hierarchical clustering affect the training cost (of the model)?

* Metrics: I am a little confused by the definition of the "final task-average performance" metric and could interpret it in at-least two ways. Could the authors please clarify this.

* I do not understand some of the results in Figures 3 and 7, for the capacity of Progressive Neural Nets (PGNs). In general, PGNs add one new column (copy of base model) for each task. So the capacity of the PGNs should always be a multiple of intial model capacity. The results do not indicate that.

* For the STL, the value of AOPD and MOPD suggests there is a good amount of variance when training the models. In this context, it would be helpful to know the variance associated with the other reported results as well.

* Figure 6 is somewhat misleading as it does not account for the mask parameters that also need to be learned per-task.

====================

Things that should be clarified in the paper (but did not impact the score):

* Is the attention (sigma) hard-attention or soft attention?

* Is the attention applied per task (ie one scalar value per task) or layer or neuron?

* Equation 2 seems to add a lot of complexity to the training mechanism (to correct for the weights of the previous tasks). Did the authors consider some other update/corrective mechanism that could be applied once a task has been learned? Please note that I am not criticizing the equation because it is complex. I am curious about the alternatives that the authors considered.

* Are there any kind of mathematical guarantees when using equation 2? If not, why should it be a better alternative to approaches like GEM[0]?

* Did the authors consider Piggyback like network for the remaining tasks as well?

====================

Certain important citations seem to be missing:

* Works like GEM[0] and AGEM[1] fix the problem of "unidirectional" transfer of knowledge to some extent.

====================

Some minor corrections for the updated version:

* Typos: eg "catastropihc",
* In the section on Large-scale training, the STL model uses 100 times more params and not 10 times.

====================

References:

[0]: GEM: https://arxiv.org/abs/1706.08840
[1]: A-GEM: https://openreview.net/pdf?id=Hkf2_sC5FX
[2]: EWC: https://arxiv.org/pdf/1612.00796.pdf



**Experience Assessment:**

I have published one or two papers in this area.

**Review Assessment: Checking Correctness Of Derivations And Theory:**

I carefully checked the derivations and theory.

**Review Assessment: Checking Correctness Of Experiments:**

I carefully checked the experiments.

**Review Assessment: Thoroughness In Paper Reading:**

I read the paper thoroughly.

---

> ### Author Response · Authors · 2019-11-12
> **Response to R1**
>
> 1) Question about Task forgetting experiments in figure 5 (d) and (e): My understanding is that after training the model on tasks 1 to 5, the weights corresponding to task 1 are dropped (that is just the task-specific parameters tau and not the mask). Then before even one gradient update is applied, the model is re-evaluated on task 1. Is that correct?
>
> -  We appreciate that you find our task forgetting experiments interesting. Since task-specific parameters also includes the mask variables, we also remove the mask M_i to completely forget the target task.
>
>
>
> 2)  Figure 7 seems to suggest that the drift is reduced by the proposed approach. What does the presence of multiple markers (of the same color) mean? For example, there are two green triangles in (b).
>
> - Each marker denotes the 2D parameters of the associated task. Since neither the task-shared nor the task-adaptive parameters for previous tasks is fixed during training in our APD formulation, we plotted multiple markers for old task-shared and task-specific parameters. For instance, we have three green markers for task 3 due to its updates when training on task 4 and 5. On the other hand, we have only a single marker for task 5, because this is the last task.
>
>
>
> 3) The choice of baselines and Comparison with A-GEM.
>
> - We appreciate your suggestion on adding in comparison against coreset-based continual learning methods. However, we believe that there is a misunderstanding, since our method, APD, does not make use of *any* previous data points as coreset-based approaches do. APD minimizes the paramter-level drift only by retroactively updating the task-adaptive parameters of the past task at each training step, without using any of the training samples from the past task. Thus we believe that our experimental validation is fair.
> However, to resolve your concern, we compared against GEM on Permuted MNIST dataset in the revision (Please see Table A.6), following all experimental results from the A-GEM paper. Table A.6 shows that APD significantly outperforms all GEM variants with minimal network capacity increase when following the experimental setup of A-GEM, although APD does not make use of *any* data instances from previous tasks. We believe that this new experimental result will further strengthen the contribution of our paper.
>
>
>
> 4) On page 2, paragraph 3, the paper mentions that "APD does not increase the intrinsic network complexity as existing expansion-based approaches do". This claim seems to be loose since a new mask is learned for each task and needs to be persisted for all the subsequent tasks. So there is an "expansion"-like step involved. The authors should clarify this detail in the context of being memory efficient.
>
> - Thanks for the helpful suggestion. What we tried to say was that APD does not change the network topology, since it makes use of additive parameter decomposition to add sparse task-adaptive parameters to the task-shared parameters, but do not introduce any new filters as existing expansion-based approaches do. We have revised the text as “APD does not increase the neurons/filters of the base neural networks”. Adding in new neurons/filters introduces lot of multiplicative operations, thus it will increase the intrinsic network complexity. APD, on the other hand, effectively handles the problem of learning richer representation for each task simply by learning (sparse) additive parameters that results in minimal capacity increase. Further, Figure 1.(b) clearly shows that APD grows the network capacity sublinearly w.r.t. the number of tasks, owing to hierarchical knowledge consolidation.
>
>
>
> 5) The authors propose a simplistic regularization approach (L2) to ensure that the shared parameters do not share too much across tasks. While it is good that a simple approach works so well, it would help if the authors discussed what do they think the reason is. EWC [2] and the authors' results indicate that regularization in the parameter space does not work as well as regularization in the function space. Thus the L2 regularization approach is not likely to work well.
>
> - As you mentioned, L2 regularization is only a simple regularization that prevents model drift. In fact, we could have used more sophisticated regularization method, but used L2 to clearly demonstrate the effectiveness of the additive parameter decomposition idea, so that we can make sure that the improvement does not come from the regularization but the parameter decomposition and updates. However, our model could use other types of regularizations, such as EWC in the leraning of task shared parameters, which we expect will achieve even higher performance than the current model.

---

> > ### Comment · AnonReviewer1 · 2019-11-14
> > **Reponse to Authors (1)**
> >
> > Thank you for taking the time to go through the comments and providing your responses.
> >
> > [Reviewer]: The choice of baselines and Comparison with A-GEM.
> >
> > [Reply from the authors]: We appreciate your suggestion on adding in comparison against coreset-based continual learning methods. However, we believe that there is a misunderstanding, since our method, APD, does not make use of *any* previous data points as coreset-based approaches do. APD minimizes the paramter-level drift only by retroactively updating the task-adaptive parameters of the past task at each training step, without using any of the training samples from the past task. Thus we believe that our experimental validation is fair.  However, to resolve your concern, we compared against GEM on Permuted MNIST dataset in the revision (Please see Table A.6), following all experimental results from the A-GEM paper. Table A.6 shows that APD significantly outperforms all GEM variants with minimal network capacity increase when following the experimental setup of A-GEM, although APD does not make use of *any* data instances from previous tasks. We believe that this new experimental result will further strengthen the contribution of our paper.
> >
> > [Reply from the reviewer]: While I agree that the proposed approach does not need *any* previous examples, I find this argument to be quite weak as GEM-like approaches require very few examples (per-task) to be stored (as also verified by follow up papers like [1]).  I accept the subsequent experiments performed by the authors.
> >
> > [1]: https://arxiv.org/abs/1811.07017
> >
> > [Reviewer]: On page 2, paragraph 3, the paper mentions that "APD does not increase the intrinsic network complexity as existing expansion-based approaches do". This claim seems to be loose since a new mask is learned for each task and needs to be persisted for all the subsequent tasks. So there is an "expansion"-like step involved. The authors should clarify this detail in the context of being memory efficient.
> >
> > [Reply from the authors]:  Thanks for the helpful suggestion. What we tried to say was that APD does not change the network topology, since it makes use of additive parameter decomposition to add sparse task-adaptive parameters to the task-shared parameters, but do not introduce any new filters as existing expansion-based approaches do. We have revised the text as “APD does not increase the neurons/filters of the base neural networks”. Adding in new neurons/filters introduces lot of multiplicative operations, thus it will increase the intrinsic network complexity. APD, on the other hand, effectively handles the problem of learning richer representation for each task simply by learning (sparse) additive parameters that results in minimal capacity increase. Further, Figure 1.(b) clearly shows that APD grows the network capacity sublinearly w.r.t. the number of tasks, owing to hierarchical knowledge consolidation.
> >
> >
> > [Reply from the reviewer]: I understand the intention of the authors. I would request them to be explicit about the capacity increase (as they are in their reply) so that the subsequent readers are not confused.

---

> > > ### Author Response · Authors · 2019-11-15
> > > **Regarding Comparison against GEM**
> > >
> > > [Reply from the reviewer]: While I agree that the proposed approach does not need *any* previous examples, I find this argument to be quite weak as GEM-like approaches require very few examples (per-task) to be stored (as also verified by follow up papers like [1]).  I accept the subsequent experiments performed by the authors.
> > >
> > > [1]: https://arxiv.org/abs/1811.07017
> > >
> > > [Reply from the authors] We will make it clear that GEM adds only a small number of examples per task. We thank you for your suggestion on the experimental validation against GEM. We believe that the experimental results against GEM have largely improved the completeness of our paper.

---

> > ### Comment · AnonReviewer1 · 2019-11-14
> > **Reponse to Authors (2)**
> >
> >
> > [Reviewer]:  Is the hierarchical clustering being done for each neuron (of the given model)? If yes, how does this approach scale to large neural nets? In general, how does the cost of doing the hierarchical clustering affect the training cost (of the model)?
> >
> > [Reply from the authors]: Hierarchical clustering of the task-adaptive parameters is done at the task-level, and thus it scales well to large neural networks. Actually, this is a main reason why our model has logarithmic capacity increase (See Figure 6, right). Hierarchical knowledge consolidation adds in negligible extra training cost to full continual learning procedure, and this is confirmed with the experimental results in Figure 3 and Table A.3, where APD(2) with hierarchical knowledge consolidation has marginally larger training time over APD(1), that does not use hierarchical knowledge consolidation.
> >
> > [Reply from the Reviewer]: I could not follow this clearly. What is the frequency of this clustering step? When the model trains on the ith task, is re-clustering performed for all the previous tasks? What is the frequency of re-clustering as the model trains on a given task?
> >
> > [Reviewer]: I do not understand some of the results in Figures 3 and 7, for the capacity of Progressive Neural Nets (PGNs). In general, PGNs add one new column (copy of base model) for each task. So the capacity of the PGNs should always be a multiple of initial model capacity. The results do not indicate that.
> >
> > [Reply from the authors] Progressive Neural Networks can add k neurons at each arrival of a new task, and we use exactly the same experimental setup for PGN as in prior expansion-based continual learning literatures (Yoon et al. 18, Xu and Zhu 18).
> >
> > [Reply from the reviewer]: PGN paper [2] clearly mentions that their approach involved quadratic parameter growth.
> >
> > [2]: https://arxiv.org/pdf/1606.04671.pdf
> >
> >
> > [Reviewer]: 16) Are there any kind of mathematical guarantees when using equation 2? If not, why should it be a better alternative to approaches like GEM[0]?
> >
> > [Reply from authors]:  By introducing a novel regularization terms with additive parameter decomposition we can maintain the original solution on previous tasks without having to preserve their data instances. GEM alleviate the forgetting problem with the use of coreset (episodic memory),  but it still has several intrinsic limitations: it cannot increase the network capacity (limited representation power) and can use only a small fraction of the previous data.
> >
> > [Reply from Reviewer]: This does not really address the original concern. Are there any mathematical guarantees available? Note that I am not necessarily holding it against the paper. I just want to ensure that any contribution from the paper does not go un-noticed. In terms of issues with GEM (specifically in terms of capacity saturation), the authors should check/refer some follow up works to GEM [3].
> >
> > [1]: https://arxiv.org/abs/1811.07017
> >
> > I accept the response for the remaining questions and update my score.

---

> > > ### Author Response · Authors · 2019-11-15
> > > **Regarding Progressive Neural Networks**
> > >
> > > [Reply from the authors] Progressive Neural Networks can add k neurons at each arrival of a new task, and we use exactly the same experimental setup for PGN as in prior expansion-based continual learning literatures (Yoon et al. 18, Xu and Zhu 18).
> > >
> > > [Reply from the reviewer]: PGN paper [2] clearly mentions that their approach involved quadratic parameter growth.
> > >
> > > [2]: https://arxiv.org/pdf/1606.04671.pdf
> > >
> > > [Reply] Thank you for your comment. We see that in the original experiment of [2], PGN grows its capacity quadratically. However, we have followed the experimental setup of [Yoon et al. 18] and [Xu and Zhu 18] which expanded PGN with k neurons at each iteration. In Section B (Compressibility of Progressive Networks) of the Supplementary Material, the authors describe that one limitation of PGN is "the growth in the size of the network with added tasks", since "the number of parameters grows quadratically", and suggest that PGN could be further compressed. Thus our experimental setup is actually in favor of PGN, since PGN that expands the size with $k$ neurons/filters can add in less number of parameters than necessary, as we are mainly evaluating the accuracy over capacity.

---

> > > ### Author Response · Authors · 2019-11-15
> > > **Regarding Comparison against GEM**
> > >
> > > [Reviewer]: 16) Are there any kind of mathematical guarantees when using equation 2? If not, why should it be a better alternative to approaches like GEM[0]?
> > >
> > > [Reply from authors]:  By introducing a novel regularization terms with additive parameter decomposition we can maintain the original solution on previous tasks without having to preserve their data instances. GEM alleviate the forgetting problem with the use of coreset (episodic memory),  but it still has several intrinsic limitations: it cannot increase the network capacity (limited representation power) and use a small fraction of the previous data.
> > >
> > > [Reply from Reviewer]: This does not really address the original concern. Are there any mathematical guarantees available? Note that I am not necessarily holding it against the paper. I just want to ensure that any contribution from the paper does not go un-noticed. In terms of issues with GEM (specifically in terms of capacity saturation), the authors should check/refer some follow up works to GEM [3].
> > >
> > > [1]: https://arxiv.org/abs/1811.07017
> > >
> > > [Response] We apologize for not understanding your question correctly. We have not yet worked on mathematical guarantees on Equation (2). We will check and refer to the follow up works to GEM as you suggested. We will clarify that the main difference of APD from GEM, is that APD does not make use of any data instances from the previous tasks.

---

> > > ### Author Response · Authors · 2019-11-15
> > > **Regarding Hierarchical Knowledge Consolidation**
> > >
> > > [Reply from the authors]: Hierarchical clustering of the task-adaptive parameters is done at the task-level, and thus it scales well to large neural networks. Actually, this is a main reason why our model has logarithmic capacity increase (See Figure 6, right). Hierarchical knowledge consolidation adds in negligible extra training cost to full continual learning procedure, and this is confirmed with the experimental results in Figure 3 and Table A.3, where APD(2) with hierarchical knowledge consolidation has marginally larger training time over APD(1), that does not use hierarchical knowledge consolidation.
> > >
> > > [Reply from the Reviewer]: I could not follow this clearly. What is the frequency of this clustering step? When the model trains on the ith task, is re-clustering performed for all the previous tasks? What is the frequency of re-clustering as the model trains on a given task?
> > >
> > > [Reply from the authors]: We perform the clustering of the task-adaptive parameters at each $s$ step (Please see Algorithm 1, Line 7-13). For the actual value of $s$ used in each experiment, please see A.1. The re-clustering is done for all previous tasks but the cluster centers are initialized as the previous cluster centers and converge fast.

---

> > > ### Author Response · Authors · 2019-11-15
> > > **Thank you**
> > >
> > > Thank you so much for your quick and timely response to our rebuttal, and updates on the review score. We believe that the clarify and completeness of our paper has largely improved thanks to your detailed comments and helpful suggestions!

---

> ### Author Response · Authors · 2019-11-12
> **Response to R1**
>
> 6) Some parts of the paper needs to be reworded or clarified more. For example, since a per-task mask is being learned, there are two sets of weights being learned per task (the mask and the task-specific parameters). So there are more parameters to learn and not just the sparse task-specific parameters.
>
> - When we mentioned task-adaptive parameters, we were talking about both the mask vector and sparse task-specific parameters. However we have clarified this in the revision as you suggested.
>
>
>
> 7) As I understand, the paper uses "order robustness"  to mean avoiding "concept drift" (or "catastrophic forgetting"). I might be missing something (in plain sight) but when I think of "order robustness", the order of tasks should not matter. This is somewhat different than avoiding concept drift.
>
> - This is a critical misunderstanding. Order-robustness is not the same as catastrophic forgetting, but is a novel problem we newly introduce in our work. We observed that conventional continual learning approaches, although successfully preventing catastrophic forgetting, are highly sensitive to the order of the task sequence. We name this behavior as ‘task-order sensitivity (Please see Figure 1(c)). This is a critical task for applications where fairness across tasks is important, such as disease diagnosis.
> Order-sensitivity does not only come from catastrophic forgetting, since models with absolutely no catastrophic forgetting, such as Progressive Neural Networks, still suffer from this problem due to the unidirectional nature of knowledge transfer in continual learning (See Page 2, paragraph 1). Our additive parameter decomposition effectively deals with this issue with the use of task-shared parameters and retroactive updates of the task-adaptive parameters.
>
>
>
> 8) Is the hierarchical clustering being done for each neuron (of the given model)? If yes, how does this approach scale to large neural nets? In general, how does the cost of doing the hierarchical clustering affect the training cost (of the model)?
>
> - Hierarchical clustering of the task-adaptive parameters is done at the task-level, and thus it scales well to large neural networks. Actually, this is a main reason why our model has logarithmic capacity increase (See Figure 6, right). Hierarchical knowledge consolidation adds in negligible extra training cost to full continual learning procedure, and this is confirmed with the experimental results in Figure 3 and Table A.3, where APD(2) with hierarchical knowledge consolidation has marginally larger training time over APD(1), that does not use hierarchical knowledge consolidation.
>
>
>
> 9) Metrics: I am a little confused by the definition of the "final task-average performance" metric and could interpret it in at-least two ways. Could the authors please clarify this.
>
> - “Final task-average performance” means that the average per-task performance after all the training is done, which is the most common metric to describe the performance of continual learning methods.
>
>
>
> 10) I do not understand some of the results in Figures 3 and 7, for the capacity of Progressive Neural Nets (PGNs). In general, PGNs add one new column (copy of base model) for each task. So the capacity of the PGNs should always be a multiple of initial model capacity. The results do not indicate that.
>
> - Progressive Neural Networks can add k neurons at each arrival of a new task, and we use exactly the same experimental setup for PGN as in prior expansion-based continual learning literatures (Yoon et al. 18, Xu and Zhu 18).
>
>
>
> 11) For the STL, the value of AOPD and MOPD suggests there is a good amount of variance when training the models. In this context, it would be helpful to know the variance associated with the other reported results as well.
>
> - Thanks for the helpful suggestion. We updated the revision with standard errors (Please see Figure 6 and Table A.5), and we observe that the performance gain of APD over existing work is significant, although APD uses significantly less number of parameters.
>
>
>
> 12) Figure 6 is somewhat misleading as it does not account for the mask parameters that also need to be learned per-task.
>
> - This is a misunderstanding. We do consider the number of all parameters, including the mask vectors, in Figure 6. Thus Figure 6 is accurate in terms of number of parameters.
>
>
>
> 13) Is the attention (sigma) hard-attention or soft attention?
>
>  - We use a soft-attention mechanism with sigmoid function (Please see page 4, ‘Additive Parameter Decomposition’ paragraph).
>
>
>
> 14) Is the attention applied per task (ie one scalar value per task) or layer or neuron?
>
> - It is a neuron-wise attention.

---

> ### Author Response · Authors · 2019-11-12
> **Response to R1**
>
> 15) Equation 2 seems to add a lot of complexity to the training mechanism (to correct for the weights of the previous tasks). Did the authors consider some other update/corrective mechanism that could be applied once a task has been learned? Please note that I am not criticizing the equation because it is complex. I am curious about the alternatives that the authors considered.
>
> - Thanks for the helpful suggestion, but as shown in the experimental results on training time (Figure 3, Table A.3), our update algorithm is already very fast compared to existing expansion-based continual learning approaches. The objective only requires regularization terms for training without multiple steps of training in Dynamically Expandable Networks, or expensive reinforcement learning in Reinforced Continual Learning.
>
>
>
> 16) Are there any kind of mathematical guarantees when using equation 2? If not, why should it be a better alternative to approaches like GEM[0]?
>
> -  By introducing a novel regularization terms with additive parameter decomposition we can maintain the original solution on previous tasks without having to preserve their data instances. GEM alleviate the forgetting problem with the use of coreset (episodic memory),  but it still has several intrinsic limitations: it cannot increase the network capacity (limited representation power) and can use only a small fraction of the previous data.
>
>
>
> 17) Works like GEM[0] and AGEM[1] fix the problem of "unidirectional" transfer of knowledge to some extent.
>
> - Thank you for the helpful suggestion. We included in discussions about episodic memory/coreset-based continual learning approaches on related work section, although we were mainly comparing against standard continual learning approaches that cannot access samples for the previous tasks.
>
>
>
> 18) Typos and corrections
>
> - Thanks for your suggestion, we have corrected them in the revision.

---

### Official Review · AnonReviewer2 · 2019-10-23
**Official Blind Review #2**

**Rating:** 1

**Review:**

Summary: The paper addresses continual learning challenges such as catastrophic forgetting and task-order robustness by introducing a new hybrid algorithm that uses architecture growth as well as parameter regularization where parameters of each layer are decomposed into task-specific and task-private parameters. They also use a simple trick to The authors perform experiments on Split CIFAR100, CIFAR100 Superclass, Omniglot, and a sequence of 3 datasets (SVHN,CIFAR10,CIFAR100). The maximum number of tasks in the experiments is 100 for Omniglot-rotation. The authors show superior performance to EWC (a regularization-based method), P&C (architecture-based method), DEN (architecture-based method), PGN (architecture-based method), RCL (architecture-based method), etc.

Pros:
+ The paper is well-written and has motivated the problem of scalability and forgetting
+ Proposing a new hybrid approach that benefits from the best of both worlds (maximum usage of the capacity with parameter regularization followed by logarithmic architecture growth at arrival of new task using layer-wise parameter decomposition.

Cons that significantly affected my score and resulted in rejecting the paper are as follows:

1- Lack of measuring forgetting:
Authors indicate in the abstract that “a continual learning model should effectively handle catastrophic forgetting” and reiterate on this on other parts of the paper yet there is no table/figure that shows the initial performance of the model on each task so that readers can compare it with the reported accuracy after being done with all tasks. Having a method with high average accuracy does not necessarily mean it has minimum forgetting. You can use forgetting measurements such as BackWard Transfer, introduced in [1] or forgetting ratio defined in [2] for this assessment. A continual learning paper without proper measurement of forgetting is incomplete.

2 - Large-scale experiment is not convincing:
Authors believe scalability has not been addressed well in the literature (page 1&2) and claim it as one of their main contributions and making it crucial to support this claim. However, the experimental setting chosen for this claim is not convincing. Authors have chosen Omniglot-rotation as their longest sequence of tasks with 100 tasks where each task has 12 classes and in each class, there exists 80 images. This will make the total dataset of size 96K images which is still far from being large-scale. While I am aware of the fact that in the current CL literature, the maximum task sequence’s length is only 20 (Split CIFAR100) and I  agree that having an order of magnitude increase in the # of tasks is beneficial,  however, Omniglot is still a toy benchmark and does not serve as a large-scale dataset by only extending it to different random rotations. Moreover, the architecture used for this experiment is LeNet which oversimplifies the problem to address. For incremental learning, I would personally think of ImageNet as a good example and for continual learning of multiple datasets you can consider the existing sequence of 8 tasks benchmarked in [2] and [3] where you can evaluate your method on more realistic images with a total of over 400K images and significant shift in the distributions. As a side note, the key idea behind the proposed method is that this method is able to decompose the parameters into task-specific and task-private whereas in the Omniglot experiment it is not intuitive that what is there between the random rotations that is shared among the task. A more detailed discussion on this would be enlightening.

3 - No standard deviations shown in the results:
Although the results are said to be average over 3 runs, no STD is reported. Given that in the most important experiment of this paper (Omniglot) the difference between Accuracy obtained by PGN and APD is not significant (79.35% vs 81.6%).
In the current CL literature, robustness to the order of the tasks is shown by performing multiple permutations of the tasks and reporting average and STD. It is needed that authors show results for this for a fair comparison.

4 - Lack of regularization-based baselines:
Considering the fact that the proposed method is a hybrid approach, it is reasonable to compare against both architecture-based and regularization-based approaches. However, most of the baselines are chosen from the former category and EWC is the only baseline for the latter category which is relatively old and has been outperformed by large margins in the past couple of years such as SI [4], VCL[5], HAT [2], PackNet [6], MASS [7], and UCB [3].

Less major (only to help, and not necessarily part of my decision assessment):

Please consider explaining connection to prior work (HAT): While the literature review seems comprehensive, authors have missed one important previous work from ICML 2018 [2] called “Overcoming Catastrophic Forgetting with Hard Attention to the Task” or HAT. Both HAT and APD use an attention mechanism to alleviate forgetting. Considering HAT is a very strong baseline, I highly recommend authors provide a comparison with it. It’s an efficient and relatively scalable method that has very small BWT.
I recommend authors provide their method’s ability for zero-shot transfer or so called forward transfer metric to further support their method.
Hyper parameter tuning: It is also worth mentioning how the tuning process was performed. In continual learning we cannot assume that we have access to all tasks' data, hence authors might want to shed some light on this.

Minor point: On page 8, last paragraph, the authors state that a masked-based pruning technique (Piggyback) is immune to forgetting which is not an accurate statement (Note that PGN is indeed zero-forgetting by definition). All masked-based methods lose some of their performance prior to pruning. While it is correct to say that their post-pruning performance is 100% recoverable by saving the mask, forgetting should be measured with respect to their performance prior to pruning because that is their trade-off to give up accuracy in lieu of freeing space for future tasks.

References:
[1] Lopez-Paz, David, and Marc'Aurelio Ranzato. "Gradient episodic memory for continual learning." Advances in Neural Information Processing Systems. 2017.

[2] Serrà, J., Surís, D., Miron, M. & Karatzoglou, A.. (2018). Overcoming Catastrophic Forgetting with Hard Attention to the Task. Proceedings of the 35th International Conference on Machine Learning, in PMLR 80:4548-4557

[3] Ebrahimi, Sayna, et al. "Uncertainty-guided Continual Learning with Bayesian Neural Networks." arXiv preprint arXiv:1906.02425 (2019).

[4] Zenke, Friedemann, Ben Poole, and Surya Ganguli. "Continual learning through synaptic intelligence." Proceedings of the 34th International Conference on Machine Learning-Volume 70. JMLR. org, 2017.

[5] Nguyen, Cuong V., et al. "Variational continual learning." arXiv preprint arXiv:1710.10628 (2017).

[6] Mallya, Arun, and Svetlana Lazebnik. "Packnet: Adding multiple tasks to a single network by iterative pruning." Proceedings of the IEEE Conference on Computer Vision and Pattern Recognition. 2018.

[7] Aljundi, Rahaf, et al. "Memory aware synapses: Learning what (not) to forget." Proceedings of the European Conference on Computer Vision (ECCV). 2018.


---------------------------------------------------------------------------------------------------------------------------------------------------------------------
---------------------------------------------------------------------------------------------------------------------------------------------------------------------
Post-rebuttal response:

Thank you for taking the time to go through comments and providing your responses.

-----------------------------------------------------------------------------------------------------------------------------------------------------------------------------------------------------------
[Authors' response:] In Table 2 (paragraph “Continual learning with heterogeneous datasets”), we have experimental results with heterogeneous datasets, where continual learning models are evaluated on a sequence of different datasets (CIFAR-100, CIFAR-10, SVHN). We agree that experiments with massive datasets will be helpful, but we do not have sufficient time to perform all the experiments during the short rebuttal period. Hence, we only compared against HAT in our revision (Please see Table A.7 and Figure A.10) on the sequence of 8 tasks [2][3] you mentioned. We directly followed all experimental settings on the paper and the code of the authors (https://github.com/joansj/hat). APD (82.42 +- 0.5 %) outperms HAT (80.36 +- 1.2 %) in terms of accuracy. Although HAT shows a marginal forgetting (0.14 %) during training, the models is task-order sensitive (AOPD: 7.95%, MOPD: 23.15%) on difficult sequences of tasks as CIFAR10, CIFAR100, and FaceScrub (Please see Figure A.10) while APD consistently shows a reliable performance with lower OPDs (AOPD: 2.09%, MOPD: 4.40%) regardless of the task order. We will add in all baselines and APD variants in the final version of the paper for this 8-dataset experiment, if it gets accepted.

[Reviewer's response:] while I thank the authors in providing this comparison, I would not call this "significant outperforming". According to Table A.7 HAT method achieves  80.36% using the memory needed to store one single network in the memory on which they learn attention masks without using extra memory while APD uses 81% more memory only to achieve 82.42% average accuracy (2.13% increase) which is clearly not a fair comparison to me. Authors should either use the same memory for HAT (using a larger network architecture) or use a smaller memory size for APD and re-evaluate this comparison otherwise it is not conclusive which method is superior. Given the large difference in memory usage I suspect HAT will outperform ADP if given more capacity. While I agree with authors that regularization based approaches can be limited by the number of tasks, in the experimental setting used in this paper, this is not proven to be the case as these methods have not reached their maximum capacity and in fact are still performing strongly well compared to a hybrid approach which is presumably supposed to be better. I appreciate the novelty of the idea of decomposing parameters but it is not clear whether this factorization is actually performed given the high capacity needed to learn these tasks. Therefore, the results are still not convincing to me.
In addition,  as also brought up by R1, "My biggest concern is the choice of baselines. The paper (rightly)  highlights that their work improves over many existing works as they provide a mechanism to "retroactively update task-adaptive parameters" for the previous task. But none of their baselines have this mechanism built-in. So while there is a clear advantage with the proposed approach, the comparison is unfair and the baselines should have considered approaches like GEM [0[ and A-GEM[1] while also provide a kind-of retrospective mechanism to correct the weights corresponding to the subsequent tasks. Without such a comparison, it is difficult to comment on the benefits of the approach." --> regarding the comparison between APD and HAT in the order-robustness, this also seems as a big concern to me. Maybe I am missing it in the long list of comments and replies, but I am not able to find a clear response from authors in providing a fair assessment without their order robustness constraint mechanism (Eq. 2). This is an auxiliary advantage that only APD is benefitting from and makes the comparison difficult.  It should be either given to all methods or none.

-----------------------------------------------------------------------------------------------------------------------------------------------------------------------------------------------------------
[Authors' response:] This is a critical misunderstanding as those results are already provided in the paper.  We already clearly report the performance evolution of several tasks during the course of training in Figure 5 (a)-(c). Figure 5 (a)-(c) clearly show that APD does not suffer from catastrophic forgetting and even improves performance on previously trained task during continual learning (Figure 5 (b)), which is an effect of update on the task-shared parameters. Please see Page 7, “Preventing catastrophic forgetting” paragraph for more detailed discussions.

[Reviewer's response:] I disagree with authors' response regarding Figure 5 being a complete forgetting measurement supported by this sentence in “Preventing catastrophic forgetting” paragraph: "APD-Nets do not show any sign of catastrophic forgetting" and showing the performance of only 3 tasks out of 20 in which accuracies are barely readable due to the coarse scale of the figure and more importantly authors do not provide other methods' performance on these tasks. However, I thank authors for providing the comparison with stronger baselines and proper forgetting measurement in A.6 and A.7, I highly recommend swapping Figure 5 with your newly obtained quantitative forgetting measurements shown in A.6 and A.7 in the appendix
 (once fairly compared according to my comment above) as they provide a better support for forgetting avoidance.

-----------------------------------------------------------------------------------------------------------------------------------------------------------------------------------------------------------
[Authors' response:] The only hyperparameters required for APD are \lambda_1 and \lambda_2, which controls the capacity of the task-adaptive parameters and model drift respectively.  Since there is a trade-off between efficiency (network capacity) and accuracy, the users only need to tune them according to their priority. The model is not sensitive to hyperparameter configurations unless they are in the correct scale, and the details of the hyperparameter configurations are given in A.1.

[Reviewer's response:] Hyper-parameters can be a lot more than just \lambda_1 and \lambda_2 in your experiments. Batch size, optimizer's learning rate, weight decay, validation set size, etc are the parameters that are usually left out  in the CL settings and are not explained how they were tuned. As I said before, it is worth explaining what this sentence from A.1 means "All hyperparameters are determined from a validation set." so if this validation is composed of the data from all tasks or they followed some procedure like A-GEM paper in which it is assumed data of only 3 tasks is available in the beginning for tuning purposes.
--------------------------------------------------------------------------------------------------------------------------------------------------------------------------------------------------------------
[Authors' response:]  What you mentioned about mask-based methods may be true in general, but not in the case of Piggyback. Piggyback is composed of frozen backbone and task-specific masks, such that the backbone network is fixed without any updates during training, and the model only learns the task-specific pruning masks, which are *stored*, such that we can recover the performance on any previous tasks at any future points. Thus Piggyback does not perform actual pruning, and thus there is no loss of accuracy and forgetting on the previous tasks.

[Reviewer's response:] I disagree with your statement about "no loss of accuracy". Restating from Piggyback paper on its page 4: "The key idea behind our method is to learn to selectively mask the fixed weights of a base network, so as to improve performance on a new task. We achieve this by maintaining a set of real-valued weights that are passed through a deterministic thresholding function to obtain binary masks, that are then applied to existing weights. By updating the real-valued weights through backpropagation, we hope to learn binary masks appropriate for the task at hand."
This simply means they learn binary mask per task by using a thresholding function and save this mask. However, if you evaluate the model on a given task **prior to making** you obtain a different performance compared to evaluating after masking (this is what you save) where the former is usually higher or sometimes similar to the former because prior to masking, there exist more parameters while by masking some parameters will be 'freed' to be used for future tasks. This difference is what I am referring to as true forgetting and is zero only if evaluation prior and post masking are exactly the same because by storing the learned masks you can only recover the post masking performance.
-----------------------------------------------------------------------------------------------------------------------------------------------------------------------------------------------------------

[Authors' response:] The scalability problem we aim to tackle in our work is the scalability to number of tasks. This is because scalability to the size of the network or the number of data instances is basically the problem with generic machine learning and are not the main problem associated with continual learning. Since continual learning models learns on a sequence of tasks, we were more interested in how the existing (expansion-based) continual learning methods and ours behave on large number of tasks, in terms of catastrophic forgetting and network capacity. However, we agree that the term ‘large-scale continual learning’ may be misleading and have renamed the paragraph to ‘scalability to large number of tasks’ in the revision.

[Reviewer's response:] I disagree with the first sentence that dataset size is not the main problem associated with CL. It is an important factor that should be considered because dataset size and number/diversity of classes can significantly increase forgetting on early tasks as the distribution shift between the tasks will be significant. I understand the intention of authors and their interest in modeling large sequence of tasks, however introducing the method as scalable is misleading and in addition to the text which is corrected now, should be also corrected in the title of the paper.

-----------------------------------------------------------------------------------------------------------------------------------------------------------------------------------------------------------
I accept the response for the remaining questions from authors but intend to keep my score. However, I will be happy to update it based on the authors' response to the very first comment above regarding providing a fair comparison in memory size and order-robustness mechanism.

**Experience Assessment:**

I have published one or two papers in this area.

**Review Assessment: Checking Correctness Of Derivations And Theory:**

I assessed the sensibility of the derivations and theory.

**Review Assessment: Checking Correctness Of Experiments:**

I carefully checked the experiments.

**Review Assessment: Thoroughness In Paper Reading:**

I read the paper thoroughly.

---

> ### Author Response · Authors · 2019-11-13
> **Response regarding Lack of forgetting measure and large-scale experiments.**
>
> Thank you for your long and detailed comments. We respond to your comments below, which we believe will deal away with your misunderstandings and concerns.
>
> 1. Lack of measuring forgetting:
> Authors indicate in the abstract that “a continual learning model should effectively handle catastrophic forgetting” and reiterate on this on other parts of the paper yet there is no table/figure that shows the initial performance of the model on each task … A  continual learning paper without proper measurement of forgetting is incomplete.
>
> - This is a critical misunderstanding as those results are already provided in the paper.  We already clearly report the performance evolution of several tasks during the course of training in Figure 5 (a)-(c). Figure 5 (a)-(c) clearly show that APD does not suffer from catastrophic forgetting and even improves performance on previously trained task during continual learning (Figure 5 (b)), which is an effect of update on the task-shared parameters. Please see Page 7, “Preventing catastrophic forgetting” paragraph for more detailed discussions.
>
> - Furthermore, in the revision, we included in precise forgetting measures as you mentioned (average / worst case forgetting) in Table A.6. The results show that APD outperforms GEM (Gradient Episodic Memory) variants both in terms of accuracy and forgetting.
>
>
> 2 - Large-scale experiment is not convincing:
>
> 2.1) Omniglot is still a toy benchmark and does not serve as a large-scale dataset by only extending it to different random rotations. Moreover, the architecture used for this experiment is LeNet which oversimplifies the problem to address.
>
> - The scalability problem we aim to tackle in our work is the scalability to number of tasks. This is because scalability to the size of the network or the number of data instances is basically the problem with generic machine learning and are not the main problem associated with continual learning. Since continual learning models learns on a sequence of tasks, we were more interested in how the existing (expansion-based) continual learning methods and ours behave on large number of tasks, in terms of catastrophic forgetting and network capacity. However, we agree that the term ‘large-scale continual learning’ may be misleading and have renamed the paragraph to ‘scalability to large number of tasks’ in the revision.
>
> - Regarding your concern on the simplicity of the network architecture, we also report APD-Net’s performance with VGG networks in Table 2.
>
> 2.2) For incremental learning, I would personally think of ImageNet as a good example and for continual learning of multiple datasets you can consider the existing sequence of 8 tasks benchmarked in [2] and [3].
>
> - In Table 2 (paragraph “Continual learning with heterogeneous datasets”), we have experimental results with heterogeneous datasets, where continual learning models are evaluated on a sequence of different datasets (CIFAR-100, CIFAR-10, SVHN). We agree that experiments with massive datasets will be helpful, but we do not have sufficient time to perform all the experiments during the short rebuttal period. Hence, we only compared against HAT in our revision (Please see Table A.7 and Figure A.10) on the sequence of 8 tasks [2][3] you mentioned. We directly followed all experimental settings on the paper and the code of the authors (https://github.com/joansj/hat). APD (82.42 +- 0.5 %) outperms HAT (80.36 +- 1.2 %) in terms of accuracy. Although HAT shows a marginal forgetting (0.14 %) during training, the models is task-order sensitive (AOPD: 7.95%, MOPD: 23.15%) on difficult sequences of tasks as CIFAR10, CIFAR100, and FaceScrub (Please see Figure A.10) while APD consistently shows a reliable performance with lower OPDs (AOPD: 2.09%, MOPD: 4.40%) regardless of the task order. We will add in all baselines and APD variants in the final version of the paper for this 8-dataset experiment, if it gets accepted.
>
> 2.3) As a side note, the key idea behind the proposed method is that this method is able to decompose the parameters into task-specific and task-private whereas in the Omniglot experiment it is not intuitive that what is there between the random rotations that is shared among the task. A more detailed discussion on this would be enlightening.
>
> - Regardless of random rotations, tasks could share specific features such as circles, curves, and straight lines. [Gidaris et al. 18] showed that we can learn generic representations even with rotated images, where they proposed a popular self-supervised learning technique where they train the model to predict the rotation angle of randomly rotated images. We have included this discussion in the revision.
>
> Gidaris, Spyros, Praveer Singh, and Nikos Komodakis. "Unsupervised representation learning by predicting image rotations." ICLR (2018).

---

> ### Author Response · Authors · 2019-11-13
> **Response regarding other comments.**
>
> 3. No standard deviations shown in the results: Although the results are said to be average over 3 runs, no STD is reported. Given that in the most important experiment of this paper (Omniglot) the difference between Accuracy obtained by PGN and APD is not significant (79.35% vs 81.6%).
>
> - First note that PGN uses significantly more number of parameters (1,543%) than APD-Net (943%). We originally had standard errors but omitted them due to space limitation. We included them back to the revision (Please see Figure 6, Table A.5), which shows that the APD-Net (81.6 +- 0.53) significantly outperforms PGN (79.35+-0.12) with statistically significant performance gains, although APD-Net uses less than two thirds of the parameters PGN uses. In all other experimental results we report, APD-Net obtains statistically significant gains over all baselines, while using much less number of parameters over existing expansion-based continual learning methods.
>
> 4 - Lack of regularization-based baselines: Most of the baselines are chosen from the architecture-based methods and EWC is the only baseline for the regularization-based method which is relatively old and has been outperformed by large margins in the past couple of years such as SI [4], VCL[5], HAT [2], PackNet [6], MASS [7], and UCB [3].
>
>  - Our approach is fundamentally an architecture-based approach which adaptively increases the network capacity as necessary. Also, it is well known that regularization-based approaches with fixed-sized networks inevitably have limitations in maintaining the performance without forgetting, when training on a large sequence of tasks. Also, EWC is not the only regularization-based continual learning approach we compare against. We also compare against Progress & Compress (ICML 2018), and Piggyback (ECCV 2018). Piggyback is an improved version of PackNet [6] you mentioned and is very similar to HAT [2]. In Table 2, we show that APD largely outperforms Piggyback both in terms of accuracy and order-robustness. Furthermore, we added in new experimental results comparing APD with HAT on 8 different tasks in our revision (Please see Table A.7 and Figure A.10), which we also significantly outperforms.
>
> 5. Please consider explaining connection to prior work (HAT) and provide provide a comparison with it.
>
> - We included the discussion of HAT in the related work. HAT is a strong continual learning approach, but cannot expand the network capacity and performs unidirectional knowledge transfer thus suffers from the capacity limitation and order-sensitivity. On the other hand, our APD adaptively increases the network capacity by introducing task-adaptive parameters which learns task-specific features not captured in the task-shared parameters. Therefore, APD can learn richer representations compared to fixed-capacity continual learning approaches. APD also exhibit several unique properties, such as task-order robustness and trivia task forgetting. The robustness to the task order sequence is an important problem that has not been tackled by the previous approaches, and we believe that this contribution alone makes our work sufficiently novel.
>
> 6. Hyper parameter tuning: It is also worth mentioning how the tuning process was performed. In continual learning we cannot assume that we have access to all tasks' data, hence authors might want to shed some light on this.
>
> - The only hyperparameters required for APD are \lambda_1 and \lambda_2, which controls the capacity of the task-adaptive parameters and model drift respectively.  Since there is a trade-off between efficiency (network capacity) and accuracy, the users only need to tune them according to their priority. The model is not sensitive to hyperparameter configurations unless they are in the correct scale, and the details of the hyperparameter configurations are given in A.1.
>
> 7. Minor point: On page 8, last paragraph, the authors state that a masked-based pruning technique (Piggyback) is immune to forgetting which is not an accurate statement (Note that PGN is indeed zero-forgetting by definition). All masked-based methods lose some of their performance prior to pruning.
>
> - What you mentioned about mask-based methods may be true in general, but not in the case of Piggyback. Piggyback is composed of frozen backbone and task-specific masks, such that the backbone network is fixed without any updates during training, and the model only learns the task-specific pruning masks, which are *stored*, such that we can recover the performance on any previous tasks at any future points. Thus Piggyback does not perform actual pruning, and thus there is no loss of accuracy and forgetting on the previous tasks.

---

### Official Review · AnonReviewer3 · 2019-10-27
**Official Blind Review #3**

**Rating:** 8

**Review:**

The paper introduces an additive parameter decomposition (APD) approach to continual learning in the sequential task classification setting and evaluates it across a number of dimensions, including task order robustness, which is comparatively less well researched. Extensive experiments show the novel approach has superior performance to relevant baselines, and provides important data about the order robustness of popular existing approaches.

Pros:
- Paper tackles the task-order sensitivity challenge in continual learning and introduces an effective order-robust approach.
- Method is scalable, parameter growth is logarithmic, forgetting of irrelevant knowledge comes for free.
- Baselines are relevant, although they only cover one of the families of approaches. Performance looks consistently better than baselines both in terms of classification accuracy as well as order robustness.
- Although non-standard, the Omniglot experiment with 100 tasks is interesting w.r.t. both measures. Performance is on par with STL, also in terms of robustness to order.
- Order robustness measures are introduced and motivated. This is particularly relevant for future research beyond simple accuracy comparisons.

Cons:
- Network architecture used for experiments as well as the exact details of the datasets are non-standard, making results very hard to compare with other papers, so one needs to rely on provided baselines only. Please post citations for papers where the experimental methodologies were adopted from if this is not the case, I many not be familiar with them!
- Classification accuracies seems relatively low across the board, especially for CIFAR-100 results. Could you please report some results in the experimental setting used by one of your baselines in the original paper?

I am inclined to recommend acceptance due to novelty of order robustness analyses and competitive properties of the method, but I would like clarifications to my experimental questions.

**Experience Assessment:**

I have published one or two papers in this area.

**Review Assessment: Checking Correctness Of Derivations And Theory:**

N/A

**Review Assessment: Checking Correctness Of Experiments:**

I assessed the sensibility of the experiments.

**Review Assessment: Thoroughness In Paper Reading:**

I read the paper thoroughly.

---

> ### Author Response · Authors · 2019-11-12
> **Response to R3**
>
> Thank you for your constructive comments. We respond to your comments below:
>
> 1) Network architecture used for experiments as well as the exact details of the datasets are non-standard.
>
> - The base network architectures and the datasets (CIFAR-100 Split and CIFAR-100 Superclass) we used follow the settings of [Yoon et al. 18] and [Xu and Zhu 18], as these expansion-based baselines are our direct competitors. In addition to the two, we newly introduced Omniglot-rotation dataset to test the scalability of our continual learning algorithm, since such a large-scale continual learning dataset (in terms of number of tasks) did not exist.  We have clarified these points in the revision.
>
> [Yoon et al. 18] Jaehong Yoon, Eunho Yang, Jeongtae Lee, and Sung Ju Hwang. Lifelong learning with dynamically expandable networks. ICLR, 2018.
> [Xu and Zhu 18] Ju Xu and Zhanxing Zhu. Reinforced continual learning. NIPS, 2018.
>
> 2) Classification accuracies seem relatively low across the board.
>
> - We performed all experiments without data preprocessing techniques, which is the main reason why the baseline performances are relatively lower than the reported (best) performances. We have clarified this in the revision (Please see A.1).

---

> > ### Comment · AnonReviewer3 · 2019-11-15
> > **I confidently maintain my score and recommend acceptance**
> >
> > I would like to thank the authors for the clarifications, which I accept, and I welcome their proficient use of openreview!
> >
> > I believe the other reviewers have made considerable contributions to strengthen different important bits of the paper, a fine example of openreview at its best! If anything, this reinforces my confidence in the score I have assigned. However, I have primarily based my evaluation on the novelty of the order robustness evaluations, which generalize the common argument in the CL literature about forgetting early tasks vs still learning later tasks. I believe the experimental results provided sufficient support for this argument, especially in the context of recent work.
> >
> > While I agree with some of the issues kindly brought up by the other reviewers, and I thank them for diligently working with the authors to correct many of them, I need to point out that over 50% of the recently published CL works may not meet some of the criteria mentioned. While I also aspire towards stronger benchmarks, I would like to advise against discouraging excellent work, which makes clear contributions, on the basis of not meeting our aspirations.
> >
> > Congratulations on a strong submission! I believe accepting this paper will lead to raising the standards for following works by adding a (largely) overlooked direction of evaluation.

---

> > > ### Author Response · Authors · 2019-11-15
> > > **Thank you**
> > >
> > > We thank you so much for appreciating the novelty and contributions of our work! During the rebuttal period, we have done our best to faithfully address the comments from the reviewers, and believe that we have clarified the most of the concerns raised by the reviewers with additional experimental results. Thank you for your time and effort in providing us insightful and constructive comments, which we believe have significantly strengthened our paper.

---

### Author Response · Authors · 2019-11-13
**Summary of the updates in the revision**

We thank all reviewers for their constructive comments. During the rebuttal period we revised the paper to faithfully reflect the comments from all the reviewers, performing multiple sets of experiments. We have made the following updates to the revision.

1. We included a large-scale experiment (in terms of the size of the datasets) on the sequence of 8 different datasets as suggested by R2 (Table A.7 and Figure A.10).

2. We included in the experimental results against coreset-based methods (GEM, A-GEM) suggested by R1 (Table A.6), and HAT suggested by R2. The results show that our method significantly outperforms both methods, both in terms of accuracy and order-robustness

3. We included in the forgetting measures for our model and regularization / coreset-based methods, HAT, GEM, and A-GEM.

4. We included discussions on HAT, GEM, and A-GEM in the Related Work section.

5. We included the standard error to all the tables, to show that the performance gains using our method are statistically significant.

6. We corrected typos.

We believe that these new experimental results against an extensive set of baselines, along with our response to each reviewer, effectively addresses the concerns and misunderstandings of the reviewers.

We strongly believe that the problem we tackle (scalability to number of tasks, order-robustness) and the idea of additive parameter decomposition we propose to tackle the problem are both highly novel, and provide important contributions to the research in continual learning, and the new experimental results we include further strengthens the paper.

---

### Decision · Program_Chairs · 2019-12-19

**Decision:**

Accept (Poster)

**Comment:**

The submission addresses the problem of continual learning with large numbers of tasks and variable task ordering and proposes a parameter decomposition approach such that part of the parameters are task-adaptive and some are task-shared. The validation is on omniglot and other benchmarks.

The reviews were mixed on this paper, but most reviewers were favorably impressed with the problem setup, the scalability of the method, and the results. The baselines were limited but acceptable. The recommendation is to accept this paper, but the authors are advised to address all the points in the reviews in their final revision.